# Cytomegalovirus restricts ICOSL expression on antigen-presenting cells disabling T cell co-stimulation and contributing to immune evasion

Guillem Angulo[1†], Jelena Zeleznjak[2,3†], Pablo Martínez-Vicente[1,4], Joan Puñet-Ortiz[1,4], Hartmut Hengel[5,6], Martin Messerle[7], Annette Oxenius[8], Stipan Jonjic[2,3], Astrid Krmpotić[3], Pablo Engel[1,4], Ana Angulo[1,4*]

[1]Immunology Unit, Department of Biomedical Sciences, Faculty of Medicine and Health Sciences, University of Barcelona, Barcelona, Spain; [2]Center for Proteomics, Faculty of Medicine, University of Rijeka, Rijeka, Croatia; [3]Department of Histology and Embryology, Faculty of Medicine, University of Rijeka, Rijeka, Croatia; [4]Institut d'Investigacions Biomèdiques August Pi i Sunyer, Barcelona, Spain; [5]Institute of Virology, University Medical Center, Albert-Ludwigs-University Freiburg, Freiburg, Germany; [6]Faculty of Medicine, Albert-Ludwigs-University Freiburg, Freiburg, Germany; [7]Institute of Virology, Hannover Medical School, Hannover, Germany; [8]Institute of Microbiology, Department of Biology, ETH Zürich, Zürich, Switzerland

*For correspondence: aangulo@ub.edu

†These authors contributed equally to this work

**Abstract** Viral infections are controlled, and very often cleared, by activated T lymphocytes. The inducible co-stimulator (ICOS) mediates its functions by binding to its ligand ICOSL, enhancing T-cell activation and optimal germinal center (GC) formation. Here, we show that ICOSL is heavily downmodulated during infection of antigen-presenting cells by different herpesviruses. We found that, in murine cytomegalovirus (MCMV), the immunoevasin m138/fcr-1 physically interacts with ICOSL, impeding its maturation and promoting its lysosomal degradation. This viral protein counteracts T-cell responses, in an ICOS-dependent manner, and limits virus control during the acute MCMV infection. Additionally, we report that blockade of ICOSL in MCMV-infected mice critically regulates the production of MCMV-specific antibodies due to a reduction of T follicular helper and GC B cells. Altogether, these findings reveal a novel mechanism evolved by MCMV to counteract adaptive immune surveillance, and demonstrates a role of the ICOS:ICOSL axis in the host defense against herpesviruses.

## Introduction

A robust host response against viral infections requires the activation of the adaptive arm of the immune system mediated by T lymphocytes. For effective T-cell activation, in addition to the antigen-specific signal delivered by the interaction of major histocompatibility (MHC)-peptide complexes and T-cell receptors (TCR), concerted co-stimulatory signals are essential. These signals come from a number of co-stimulatory molecules upon interaction with their cognate ligands, expressed on antigen-presenting cells (APCs) (*Sharpe, 2009*; *Chen and Flies, 2013*). The B7-CD28 family of ligands and receptors comprises molecules of the Ig superfamily that function as co-signaling molecules modulating T-cell responses (*Sharpe and Freeman, 2002*). The best characterized co-stimulatory pathway between members of this family is exemplified by the T-cell surface receptor CD28, which binds to the activated APC surface molecules B7-1 (CD80) and B7-2 (CD86). Another co-

stimulatory pathway of this family is the receptor ICOS (inducible co-stimulator, CD278), primarily expressed on activated T cells and at very high levels on T follicular helper cells (Tfh), which mediates its functions by binding to its ligand ICOSL (B7-H2, CD275) (*Hutloff et al., 1999*; *Yoshinaga et al., 1999*). ICOSL is constitutively expressed on APCs, such as dendritic cells (DCs), macrophages, and B cells, and it is strongly upregulated by inflammatory stimuli (*Aicher et al., 2000*; *Swallow et al., 1999*; *Yoshinaga et al., 2000*). ICOSL presents a different expression profile as compared to CD80 and CD86, and it is induced on non-lymphoid cells, such as endothelial cells or epithelial cells (*Khayyamian et al., 2002*; *Qian et al., 2006*). The ICOS:ICOSL receptor pathway leads to enhanced T-cell activation, proliferation, and Th2 responses (*Riley et al., 2002*; *Tesciuba et al., 2001*). In contrast to CD28, which is essential for the activation of naïve T cells, ICOS is more relevant for the regulation of activated and effector T cells (*Coyle et al., 2000*). In addition, ICOS co-stimulation plays an essential role in T-dependent antibody responses and germinal center (GC) formation, by inducing IL-21 production and Bcl6 expression, consequently controlling isotype switching, somatic hypermutation, and the generation of memory B cells and plasma cells (*Choi et al., 2011*; *Hutloff, 2015*; *Liu et al., 2015*). ICOS has also been shown to be required for optimal CD8[+] T-cell proliferation and cytokine production during recall responses (*Takahashi et al., 2009*; *Wallin et al., 2001*) and NK cell activation (*Ogasawara et al., 2002*). Importantly, although the ICOSL:ICOS axis is a crucial tuner of cell-mediated immune responses, little is known about its role in viral host defence.

Throughout the process of pathogen-host coevolution, viruses have devised multiple strategies to counteract host immunity. In particular, in order to ensure their prolonged survival in their hosts, persistent viruses need to undermine adaptive immune responses at different levels. A common theme of these viruses is to interfere with the presentation of antigenic peptides by MHC molecules to T cells. To this end, several large DNA viruses, particularly herpesviruses, encode multiple proteins, which usually are not essential for replication in vitro, with the capacity to interfere with the MHC molecules or additional cellular components involved in antigen presentation pathway (*Schuren et al., 2016*). Given the importance of APCs in triggering and maintenance of adaptive antiviral immune responses, herpesviruses have evolved molecular determinants and mechanisms to impair their functions (*Brinkmann et al., 2015*; *Gewurz et al., 2007*). Among them, different members of the herpesvirus family have been reported to disrupt the interaction established by co-signaling molecules in infected APCs and their counter receptors, in particular by downregulating CD80 and CD86. This is the case for example of the human and murine cytomegaloviruses (HCMV and MCMV, respectively), which cause the downregulation of these two molecules during their respective infectious courses (*Hertel et al., 2003*; *Loewendorf et al., 2004*; *Mintern et al., 2006*; *Moutaftsi et al., 2002*). However, whether viruses have evolved immune evasion strategies to counteract the ICOSL:ICOS pathway remains largely unexplored.

Here, we report that ICOSL-mediated co-stimulation contributes to the development of effective anti-MCMV immune responses. In addition, we demonstrate that ICOSL constitutes a major target of CMVs and other herpesviruses, which potently downregulate its expression on the surface of APCs. We show that MCMV uses its immunoevasin m138/fcr-1 to manipulate this co-signaling molecule, impeding a correct T-cell activation during acute MCMV infection. These results reveal a new mechanism of viral immune evasion.

## Results

### MCMV infection rapidly induces a potent ICOSL downregulation in APCs

ICOSL is constitutively expressed at high levels on APCs, including DCs and macrophages, which are cell types that are susceptible to MCMV infection. In order to evaluate whether MCMV has the capacity to alter cell-surface expression of ICOSL, we initially mock-infected or infected murine peritoneal macrophages, at a multiplicity of infection (moi) of 10, with a GFP expressing MCMV (MCMV-GFP). Under these conditions, around 50% of the cells in the culture were infected. Three days later, cultures were analyzed by flow cytometry to measure the expression of cell surface ICOSL. At this time point, as expected, the density of MHC I at the cell surface was strongly reduced (*Figure 1A*). Notably, a pronounced downmodulation of ICOSL was also observed in GFP+ infected cells as compared with uninfected GFP- cells from the same cultures, or with mock-infected samples

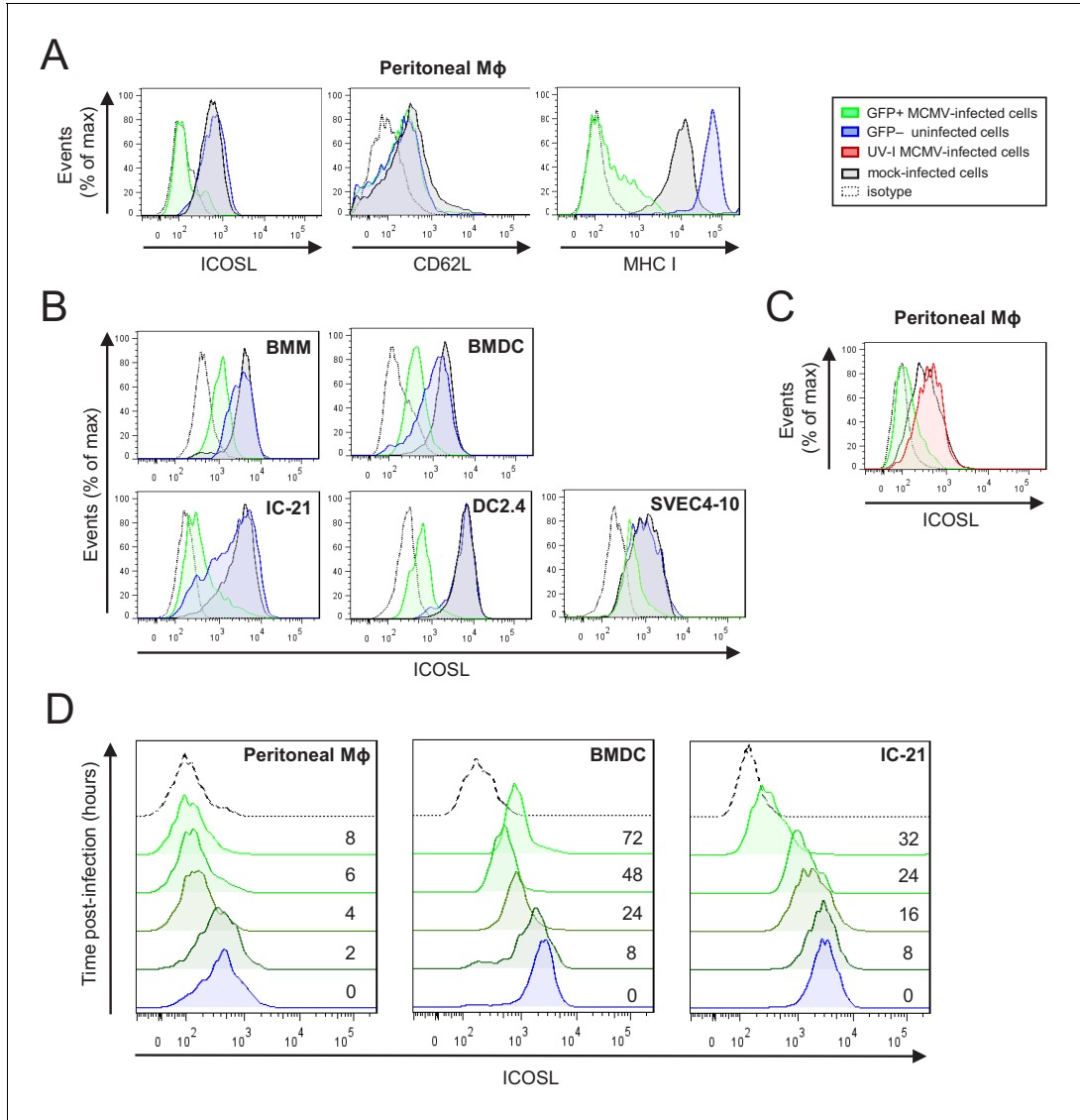

**Figure 1.** Downregulation of ICOSL upon MCMV infection of antigen-presenting cells (APCs). (**A**) Peritoneal macrophages (MΦ) were mock-infected or infected for 72 hr with MCMV-GFP at an moi of 5 and analyzed by flow cytometry for surface expression of ICOSL, MHC I, or CD62L using specific mAbs against each of these molecules. Gray histograms represent the expression on mock-infected cells, green histograms represent the expression on MCMV-infected (GFP+) cells, and blue histograms represent the expression on uninfected (GFP-) cells from the same culture. (**B**) Bone-marrow-derived macrophages (BMM), bone-marrow-derived dendritic cells (BMDC), IC-21, DC2.4, and SVEC4-10 cells were infected with MCMV-GFP at different moi (10, 20, 20, and 5, respectively) to obtain an infection of around 50% of the cell culture, and analyzed by flow cytometry for surface expression of ICOSL. Gray, green, and blue histograms, as indicated in A. (**C**) Same as in A, except that peritoneal macrophages were infected with MCMV-GFP at an moi of 10, or treated for 72 hr with the same amount of MCMV-GFP UV-inactivated. Gray and green as indicated in A, and red histograms represent the expression on MCMV-GFP UV-inactivated infected cells. (**D**) Peritoneal macrophages, BMDCs, and IC-21 cells were mock-infected (time 0) or infected with MCMV-GFP at an moi of 10, 40, and 20, respectively, and analyzed by flow cytometry for surface expression of ICOSL at the different time points after infection indicated. In **A**, **B**, **C**, and **D**, the isotype for each antibody was used as a negative control (dotted lines). Data are representative of at least two independent experiments.

(*Figure 1A*). Indeed, ICOSL levels decreased to almost undetectable levels, whereas the expression of the adhesion molecule CD62L, used as a negative control, remained unaffected. We next assessed if these findings could be extended to primary bone-marrow-derived macrophages (BMM) and dendritic cells (BMDC). Importantly, while robust expression of ICOSL was observed on all uninfected (mock or GFP-) cells, the surface reduction of this molecule in MCMV-infected BMMs and BMDCs was significant (*Figure 1B*). Similar results were also obtained upon infection of the macrophage cell line IC-21, the DC line DC2.4, and the endothelial cell line SVEC4-10, although the extent

of downregulation differed between them (*Figure 1B*), being quite dramatic on the IC-21 and DC2.4 cells, and more subtle in the SVEC4-10. Altogether, these findings demonstrate that MCMV infection causes a consistent and specific downmodulation of ICOSL from the surface of APCs.

We then assessed whether de novo viral protein synthesis was required for ICOSL downregulation. To this end, peritoneal macrophages were infected for 72 hr (infection rates, approximately 90%) with UV-inactivated MCMV, and the results showed no loss of ICOSL surface expression (*Figure 1C*), indicating that a productive MCMV infection was required to alter ICOSL expression. To determine the kinetics of ICOSL downmodulation, we examined its surface expression levels at different time points following infection of peritoneal macrophages with MCMV-GFP. As shown in *Figure 1D*, by 4 hr post-infection (hpi) the density of ICOSL at the cell surface was substantially reduced, and by 6 hpi there was barely any ICOSL at the surface of infected cells. When a similar kinetics was conducted in BMDC and IC-21 cells, the downregulation of ICOSL differed with respect to timing, beginning to be evident by 8 hpi and 16 hpi, respectively, with the effect becoming more prominent from that time point onwards. Thus, we concluded that MCMV encodes one or more proteins to rapidly lower cell surface levels of ICOSL upon infection of APCs.

## The MCMV m138 gene is responsible for ICOSL downmodulation

In order to find the viral gene/s mediating the observed effects on ICOSL expression, we screened a panel of GFP-expressing MCMV deletion mutants that lack different regions of the viral genome, and which covered a large number of genes dispensable for viral replication in cell culture. The levels of ICOSL were monitored by flow cytometry 3 days after infection of IC-21 cells with the different mutants. We found that, except for the deletion mutant MCMVΔm128-138, all MCMV mutants tested resulted in the marked downmodulation of ICOSL expression (*Figure 2A*; see as an example results for the MCMVΔm1-17 mutant). The viral region deleted in MCMVΔm128-138, expanding from ORF m128 to m138, includes some ORFs with functions still unassigned and others involved in viral-host interactions, such as the m129/m131 chemokine homolog associated with tropism to macrophages (*Fleming et al., 1999*; *Stahl et al., 2015*), or the m138 protein, originally described as an Fc receptor (*Thäle et al., 1994*), and later reported to be also implicated in the reduction of cell surface expression of the NKG2D ligands H60, MULT-1, and RAE-1ε (*Arapović et al., 2009*; *Lenac et al., 2006*), and interestingly, of the co-stimulatory molecule CD80 (*Mintern et al., 2006*), a member of the B7 family to which also ICOSL belongs. Based on this later feature of m138 and the fact that the viral protein is expressed early during infection, we speculated that the gene altering ICOSL expression in MCMV-infected cells might be m138. To directly test the hypothesis, we used an MCMV with a deletion in this gene (MCMVΔm138). As illustrated in *Figure 2B*, during infection of IC-21 cells with MCMVΔm138, cell surface expression of ICOSL was maintained to levels similar to mock-infected cells, while the levels of CD84, a molecule targeted by the MCMV m154 protein (*Strazic Geljic et al., 2020*), used as a control, markedly decreased. These effects were not due to a growth defect of the mutant virus in IC-21 cells, as indicated in earlier studies where comparable multi-step growth kinetics between an MCMV deleted in m137-m139 and parental MCMV were observed (*Cavanaugh et al., 1996*). These results revealed that one gene product, m138, is responsible for the complete downregulation of ICOSL during MCMV infection.

We next asked if m138, in the absence of other viral molecules, was sufficient to decrease cell surface levels of ICOSL. To this end, we cloned the m138 gene fused at the C-terminus with the GFP protein and used the resulting recombinant plasmid (named m138-GFP) or the parental GFP only-expressing plasmid (CTL-GFP) to transiently transfect IC-21 cells. Flow cytometry analysis 24 hr after transfection showed that cells transfected with the m138-expressing construct (GFP+) exhibited a significant reduction of ICOSL surface expression compared to the untransfected control cells from the same cultures or mock-transfected cells (GFP-; *Figure 2C*, left panel). In contrast, we could not detect any change in the expression of ICOSL in cells transfected with the GFP only-expressing plasmid (compare GFP+ and GFP- cells in *Figure 2C*, right panel). Therefore, these results indicate that m138 is able to exert its downmodulatory effects on ICOSL expression without requiring the MCMV infection context.

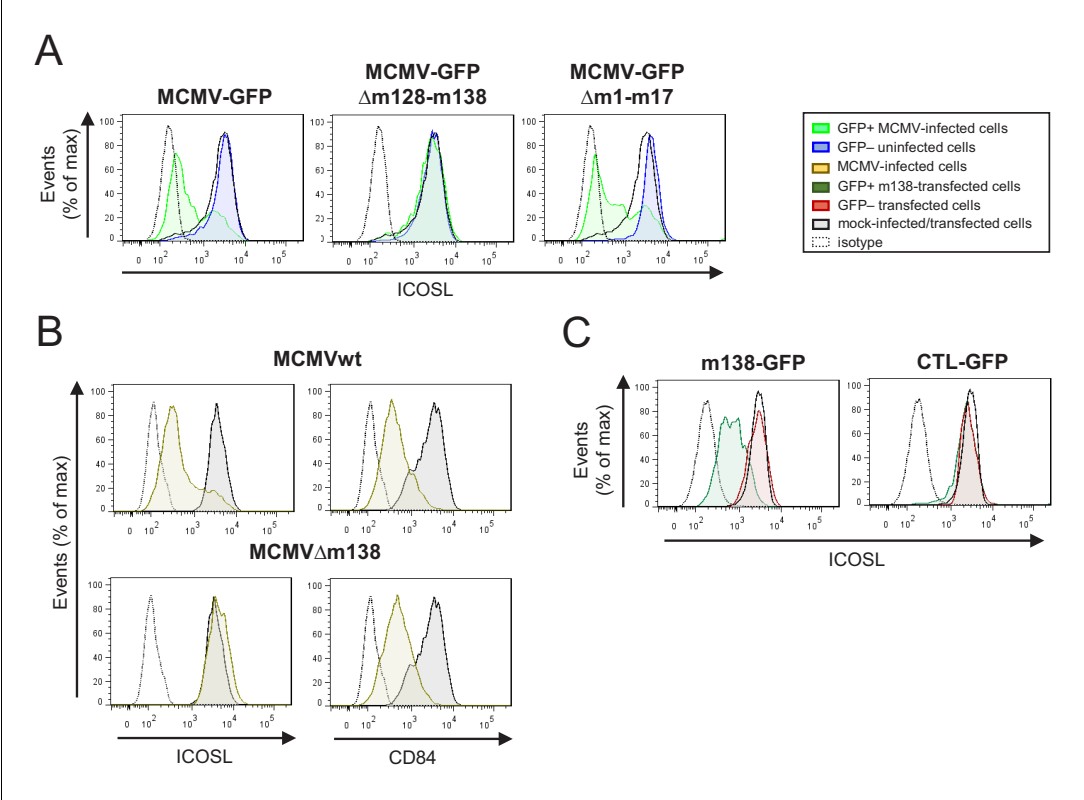

**Figure 2.** Identification of the MCMV gene involved in the decreased cell surface expression of ICOSL. (**A**) IC-21 cells were mock-infected or infected for 72 hr with MCMV-GFP, MCMV-GFPΔm128-m138, or MCMV-GFPΔm1-m17 at an moi of 10 and examined by flow cytometry for surface expression of ICOSL. Gray histograms represent the expression of mock-infected cells, green histograms represent the expression on MCMV-infected (GFP+) cells, and blue histograms represent the expression on uninfected (GFP-) cells from the same culture. (**B**) IC-21 cells were mock-infected or infected with MCMVwt or MCMVΔm138 for 72 hr at an moi of 20 and analyzed by flow cytometry for surface expression of ICOSL and CD84. Gray histograms as indicated in A, and brown histograms represent the expression on MCMVwt and MCMVΔm138 IC-21-infected cells. (**C**) IC-21 cells were mock-transfected or transfected with the m138-GFP construct or with the control GFP empty vector and stained with the anti-ICOSL mAb or an isotype control (dotted lines). Dark green and red histograms represent cells from transfected cultures expressing or not expressing GFP, respectively, and gray histograms represent surface expression of ICOSL on untransfected cells. In A, B, and C, the isotype for each antibody was used as a negative control (dotted lines). Data are representative of at least two independent experiments.

## Localization of m138 during MCMV infection

m138 has been predicted to be a 569-aa type I transmembrane glycoprotein (*Thäle et al., 1994*). Due to the lack of an m138 monoclonal antibody (mAb), the cellular localization of this viral protein during MCMV infection has so far not been directly assessed. Previous studies relied on cells expressing the m138 protein in isolation, or on the different capacity displayed by cells infected with MCMV or with an MCMV mutant lacking the viral protein to bind Ig. Thus, in order to detect m138 in the infection context, we generated a mAb specific for m138, using 300.19 cells stably transfected with the m138 protein HA-tagged at the N-terminus (HA-m138) as immunogen. Then, NIH3T3 cells were infected with MCMV-GFP, and m138 expression was analyzed by indirect immunofluorescence employing this newly generated mAb. As shown in *Figure 3A* (panels a-c and g), using permeabilizing conditions, the viral protein was detected in perinuclear compartments and cytoplasmic 'large punctate vesicles' in GFP+ infected cells. Similar results were obtained in MCMV-GFP infected DC2.4 cells (*Figure 3—figure supplement 1*). Therefore, these results were consistent with earlier reports in transfected cells, showing the m138 protein being mainly expressed in intracellular compartments together with markers of the ER, protein disulfide isomerase-A1 (PDI) and calnexin, in addition with the lysosomal marker LAMP-1 (*Mintern et al., 2006*; *Thäle et al., 1994*). In contrast, in non-permeabilized infected cells, the mAb against m138 failed to detect the viral protein at the cell surface (*Figure 3A*, panels d-f), whereas it could be recognized at this location in the 300.19 cells

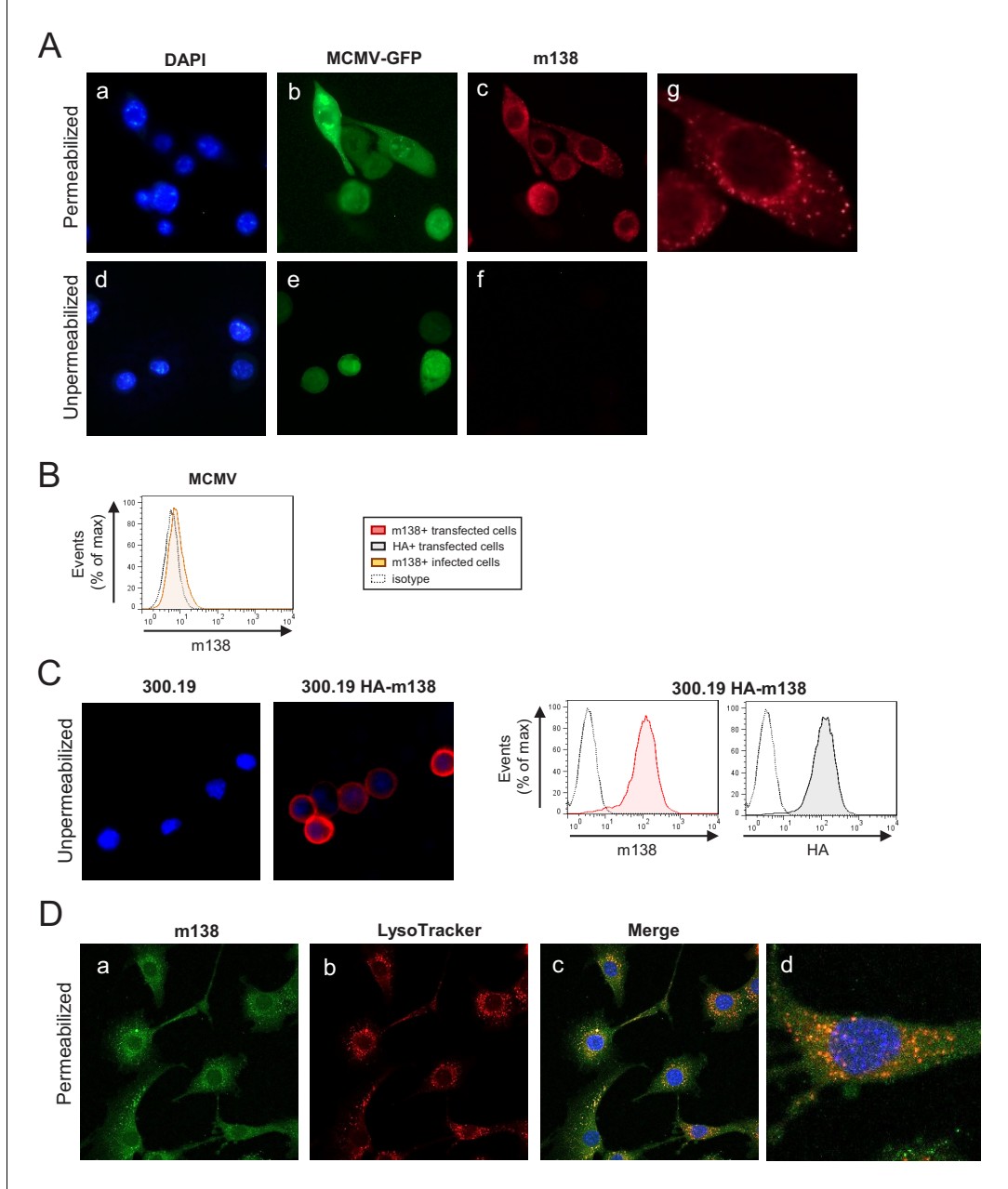

**Figure 3.** Localization of m138 in MCMV-infected cells. (**A**) NIH3T3 cells were infected with MCMV-GFP at an moi of 5 for 24 hr, fixed with 4% formaldehyde (panels d-f) or fixed and permeabilized with 0.05% Triton (panels a-c and g), and stained with the anti-m138 mAb followed by an anti-mouse IgG-A555. Nuclei were stained with the DAPI reagent. The cells were examined under a fluorescence microscope. Magnification, ×20 (a-f); an enlarged individual cell from panel c is shown in panel g. (**B**) MCMVwt-infected NIH3T3 cells were examined by flow cytometry for surface expression of m138 with the anti-m138 mAb (red histogram). Dotted line represents the isotype control. (**C**) 300.19 cells untransfected or stably transfected with the HA-m138 construct were fixed with 4% formaldehyde and stained with the anti-m138 mAb and DAPI and examined under a fluorescence microscope as indicated in A. Magnification, ×20. Overlaid images are shown. Transfected cells from the same cultures were analyzed by flow cytometry (right panels) with the anti-m138 mAb (red histogram), an anti-HA mAb (gray histogram) or isotype controls (dotted lines). (**D**) NIH3T3 cells were infected with MCMV at an moi of 5 for 24 hr, treated with LysoTracker DND99, fixed with 4% formaldehyde and permeabilized with 0.02% Saponin, and stained with the anti-m138 mAb followed by an anti-mouse IgG-A488. Nuclei were stained with the DAPI reagent. The cells were examined under a confocal microscope. Representative fields are shown in panels a, b, and c, with an enlargement of an individual and more exposed cell from panel c, in panel d. Magnification, ×63.

The online version of this article includes the following figure supplement(s) for figure 3:

**Figure supplement 1.** Localization of m138 in MCMV-infected DC2.4 cells.

stably transfected with HA-m138 (*Figure 3C*, left panels). When analyzed by flow cytometry, though, a dim expression of m138 at the cell surface of infected cells could be observed (*Figure 3B*). As expected, the viral protein was also strongly recognized at the plasma membrane of the transfected cell when assessed by flow cytometry (*Figure 3C*, right panel). Finally, to evaluate whether m138 could be found in lysosomes, NIH3T3 cells were infected with MCMVwt for 24 hr and stained with LysoTracker Red DND99, a fluorescent marker that labels acidic components and is employed to identify lysosomes. Infected cells were then analyzed by immunofluorescence using the m138 specific mAb under gently permeabilizing conditions (0.02% saponin) to preserve the LysoTracker staining. The results, illustrated in *Figure 3D* (panels c, and d), indicated that m138 colocalized with lysosomes. Indeed, quantification of the m138/lysosome colocalization, using the Coloc2 plugin from FIJI/ImageJ program, led to Manders colocalization coefficients of M1 = 0.76 and M2 = 0.80.

## m138 prevents ICOSL molecules from reaching the plasma membrane via targeting to lysosomal degradation

We next sought to study the mechanism by which m138 downregulates ICOSL. Since there are no available antibodies capable to detect murine ICOSL in western blot, we generated an NIH3T3 stable cell line expressing HA-ICOSL. We confirmed by flow cytometry that upon infection with MCMVwt, or following transient transfection with the plasmid expressing GFP-tagged m138, but not after infection with MCMVΔm138, HA-ICOSL levels specifically declined at the plasma membrane of the stable HA-ICOSL expressing cell line (*Figure 4A and B*). NIH3T3 HA-ICOSL cells were then mock-infected or infected with MCMVwt or MCMVΔm138, lysates prepared and western blots performed to examine total ICOSL protein levels by staining for the HA tag. As shown in *Figure 4C*, two ICOSL specific bands, with apparent molecular masses of ~51 and 70 kDa, were detected in the HA-ICOSL stable cell line as compared to untransfected NIH3T3 cells (compare lanes 1 and 2). Notably, in MCMV-infected cell lysates, the 70 kDa band was drastically reduced (lane 4), whereas it was not substantially altered after infection with MCMVΔm138 (lane 5), suggesting that this band corresponds to the mature surface form of ICOSL. Indeed, a mild treatment of the NIH3T3 HA-ICOSL cells with trypsin, which selectively cleaves the ectodomains of molecules at the cell surface, and therefore resulted in the loss of detection of ICOSL when assessed by flow cytometry (*Figure 4D*), led to the disappearance of the 70 kDa band in the western blot (*Figure 4C*, lane 3). Further demonstration that the larger ICOSL species corresponded to the one present at the cell surface was obtained from a parallel experiment in which we immunoprecipitated lysates from biotinylated HA-ICOSL-transfected cells with the anti-HA mAb (*Figure 4E*). Thus, m138 was altering the expression of the mature form of ICOSL that reaches the cell surface. In addition, we noted, that the 51 kDa band was more abundantly expressed in the infected samples than in the mock-infected samples (*Figure 4C*, compare lanes 2 with lanes 4 and 5). The fact that ICOSL was slightly induced at the cell surface after treatment with UV-inactivated MCMV (*Figure 1C*), could indicate that virion components or the process of viral binding may be causing these effects. Supporting this observation, an analysis of extracts from NIH3T3 HA-ICOSL cells infected with UV-inactivated MCMV indicated that this was the case, with both the 51 kDa and the 70 kDa bands being considerably more abundant than in the mock-infected samples (*Figure 4F*, compare lanes 2 and 4). Thus, MCMV infection initially induces ICOSL expression and m138 subsequently leads to the depletion of cell-surface ICOSL, but not of the overall intracellular levels of this molecule.

Taking into account the intracellular localization of m138 and the fact that it targets the B7 costimulatory molecule CD80 early in the secretory pathway, rerouting it to lysosomes, we contemplated the possibility that the viral protein was behaving in a similar way to interfere with ICOSL (*Mintern et al., 2006*). We addressed this issue by co-transfecting NIH3T3 cells with the plasmid encoding for HA-ICOSL together with either the construct expressing m138-GFP or an unrelated GFP protein (CTL-GFP), and analyzing the levels of ICOSL expression by western blot before and after treatment with a combination of two lysosomal inhibitors, leupeptin and bafilomycin. Under these transient transfection conditions, and in contrast to the assays in stably transfected NIH3T3 HA-ICOSL cells, ICOSL and m138 were simultaneously expressed, without previous existence of ICOSL at the cell surface. As shown in *Figure 5A*, in cells co-transfected with the m138-GFP protein, independently of whether or not the lysosomal inhibitor treatment was applied, the 70 kDa ICOSL band, corresponding to ICOSL on the cell surface, was not observed (lanes 4 and 5). In addition, the results also evidenced that the 51 kDa band, corresponding to intracellular ICOSL, augmented (1.5-

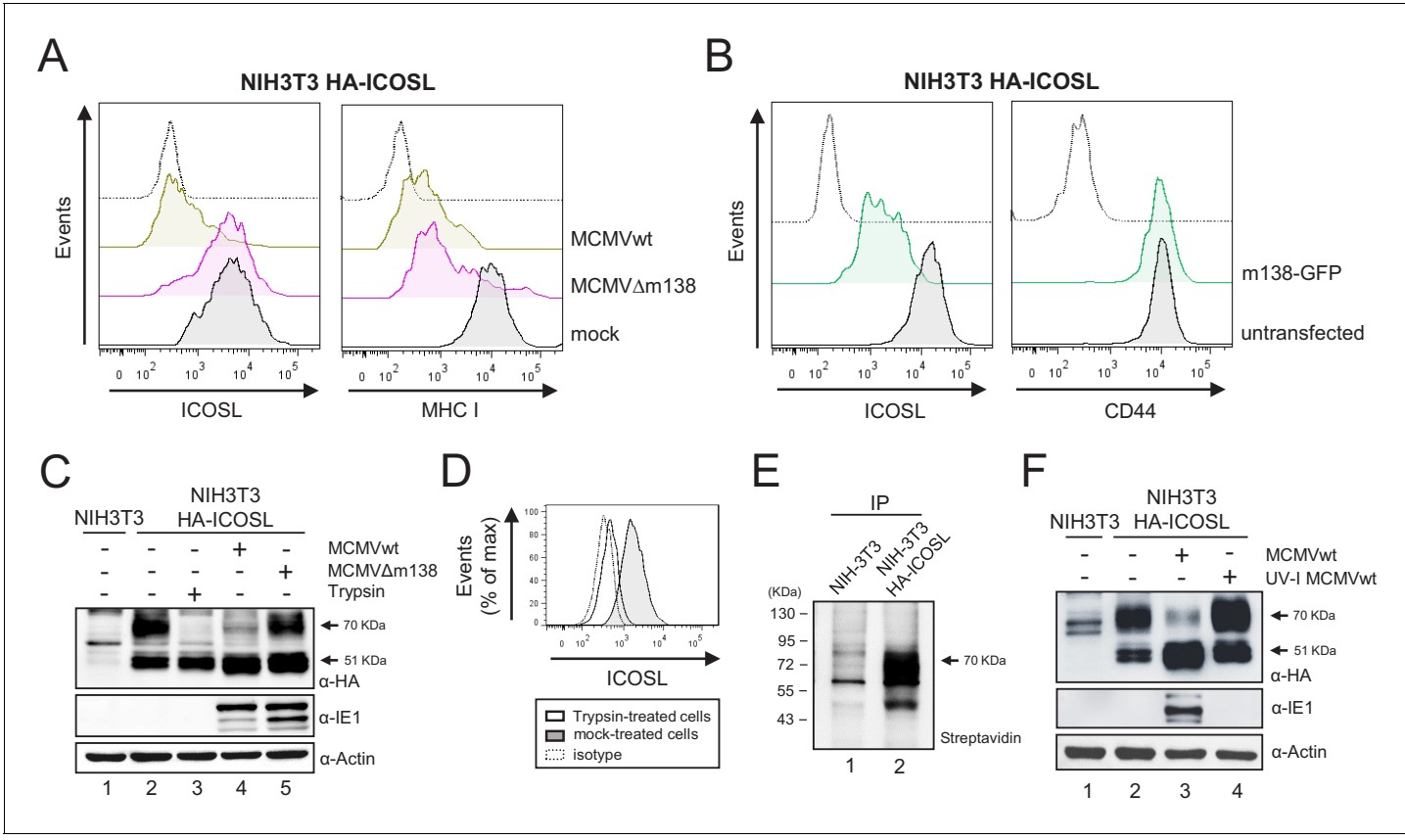

**Figure 4.** m138 leads to a reduction of the expression of ICOSL on the cell surface of infected cells but not of its overall intracellular levels. (**A**) NIH3T3 cells stably transfected with HA-ICOSL were mock-infected (gray histograms) or infected with MCMVwt or MCMVΔm138 (brown or pink histograms, respectively) and analyzed by flow cytometry for surface expression of ICOSL and MHC I. (**B**) NIH3T3 HA-ICOSL cells non-transfected (gray histograms) or transfected with the m138-GFP construct (green histograms) were analyzed by flow cytometry for surface expression of ICOSL and CD44. In A and B, the isotype for each antibody was used as a negative control (dotted lines). (**C**) NIH3T3 HA-ICOSL cells were mock-infected (lane 2), treated with trypsin (lane 3), or infected with MCMVwt (lane 4) or MCMVΔm138 (lane 5). Untransfected NIH3T3 cells were used as a control (lane 1). Samples were analyzed by western blot using an anti-HA mAb, followed by anti-rabbit IgG-HRP. An anti-IE1 and an anti-actin mAbs followed by anti-mouse IgG-HRP, were used as controls of MCMV infection and loading, respectively. (**D**) Flow cytometry analysis of NIH3T3 HA-ICOSL treated with trypsin (open histogram) or mock-treated (gray histogram) and stained with an anti-HA mAb or an isotype control (dotted line), followed by anti-mouse IgG PE. (**E**) NIH3T3 cells (lane 1) or NIH3T3 HA-ICOSL (lane 2) were surface labeled with biotin, immunoprecipitated with an anti-HA mAb, and analyzed by western blot analysis using streptavidin-HRP conjugate. (**F**) NIH3T3 HA-ICOSL cells were mock-infected (lane 2), or infected with MCMVwt (lane 3) or MCMVwt UV-inactivated (lane 4). Samples were lysed and processed as indicated in C. A lysate of untransfected NIH3T3 cells (lane 1) was also subjected to western blot and employed as a control. All infections were performed at an moi of 5 for 24 hr. Data are representative of at least two independent experiments. In **C**, **E**, and **F**, the size of the bands corresponding to ICOSL are indicated on the right margin in kilodaltons (kDa). In E, the sizes of the molecular weight markers are shown on the left side.

fold) after exposure to leupeptin and bafilomycin (compare lanes 4 and 5). In contrast, in cells expressing the CTL-GFP protein, both the 51 and 70 kDa bands were present, and the blockade of the lysosomal pathway did not lead to a substantial alteration (1.1-fold) of any of the two ICOSL forms (*Figure 5A*, compare lanes 2 and 3). The presence of the 70 kDa mature form of ICOSL when the CTL-GFP was expressed, and its absence upon expression of m138-GFP under conditions of lysosomal inhibition, pointed out that the viral protein was targeting ICOSL before reaching the cell surface. We then infected NIH3T3 HA-ICOSL cells with MCMVwt and examined by confocal fluorescence microscopy m138 and ICOSL expression. A marked colocalization of ICOSL and the viral protein in perinuclear compartments and in cytoplasmic punctate structures could be observed, being more robust after treatment with the lysosomal inhibitors (*Figure 5B*, panels g and h). Accordingly, the expression signal of ICOSL was also increased after treatment with leupeptin and bafilomycin (compare panels a and e in *Figure 5B*). The results also showed augmented levels of m138 upon exposure to the lysosomal inhibitors (compare panels b and f), indicating that m138 was getting

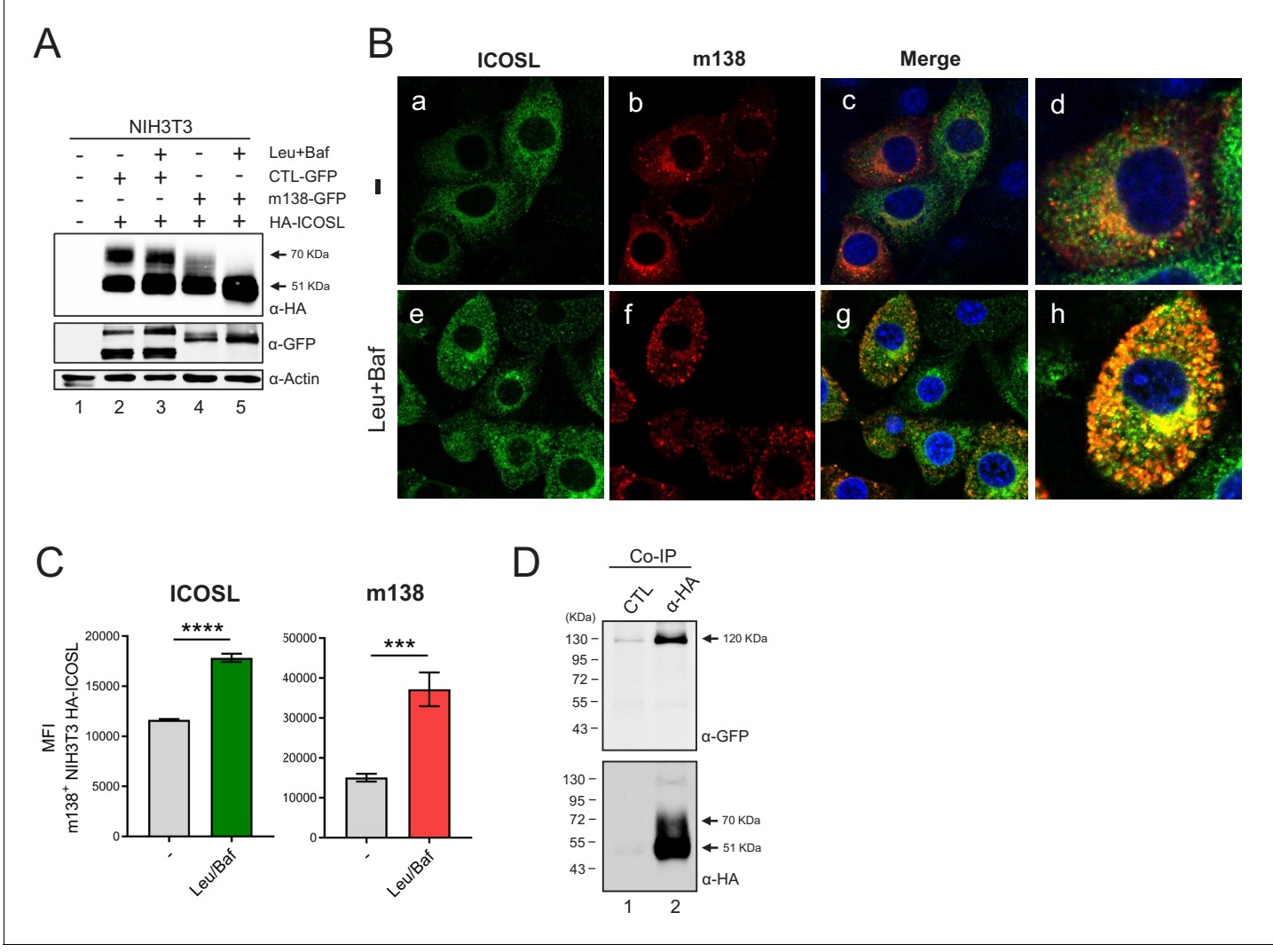

**Figure 5.** m138 impedes ICOSL maturation redirecting it to lysosomal degradation. (**A**) Western blot analysis of NIH3T3 cells untransfected (lane 1) or transiently co-transfected with either the HA-ICOSL and the control-GFP (CTL-GFP) constructs (lanes 2 and 3), or the HA-ICOSL and the m138-GFP constructs (lanes 4 and 5) and, when indicated, treated with 250 µM of leupeptin and 20 nM of bafilomycin A1 (lanes 3 and 5). The expression of ICOSL and actin was assessed as described in *Figure 4C*, and for the expression of the m138-GFP and CTL-GFP proteins a polyclonal antibody anti-GFP followed by anti-rabbit IgG-HRP were used. (**B**) NIH3T3 HA-ICOSL were infected with MCMVwt at an moi of 5 for 24 hr in the absence (-; panels a, b, c and d) or presence of leupeptin and bafilomycin (panels e, f, g and h). Cells were fixed, permeabilized, and stained with anti-m138 and anti-ICOSL mAbs followed by an anti-mouse IgG-A555 and anti-rat IgG-A488, respectively. Nuclei were stained with the DAPI reagent and samples were examined under a confocal fluorescence microscope. Shown are representative cells from cultures stained for m138 (panels a, e), ICOSL (b, f), and overlaid images (c, d, g, and h). Enlarged and more exposed individual cells from panels c and g are presented on panels d and h, respectively. Magnification, ×63. (**C**) NIH3T3 HA-ICOSL infected and treated with the lysosomal inhibitors as in B were fixed, permeabilized with the Foxp3 staining buffer set, and analyzed by flow cytometry with anti-m138 and anti-ICOSL mAbs followed by an anti-mouse IgG-A555 and anti-rat IgG-A488, respectively. Conditions were analyzed in triplicates and median fluorescence intensity (MFI) values +/- SD were extracted from pre-gated m138 positive cells. (**D**) NIH3T3 cells were co-transfected with the m138-GFP and HA-ICOSL constructs. HA-ICOSL was immunoprecipitated from 0.5% Triton-treated lysates using an anti-HA mAb or an anti-hFc as a control. Recovered immunoprecipitates were subjected to SDS-PAGE and western blot with antibodies against GFP and HA, as described in A and *Figure 4C*. Data are representative of at least two independent experiments. In A and D, the size of the two bands corresponding to ICOSL, and in D, the band corresponding to m138-GFP, are indicated on the right margin in kilodaltons (kDa). In D, the sizes of the molecular weight markers are shown on the left side.

partially co-degraded with its target. These observations were confirmed by flow cytometry in permeabilized NIH3T3 HA-ICOSL MCMV-infected cells, where significant higher levels of expression of both ICOSL and m138 could be detected after blockade of the lysosomal pathway (*Figure 5C*).

Collectively, the data suggest that m138 precludes the maturation of ICOSL by targeting it to lysosomes, explaining how the levels of this co-signaling molecule are decreased at the cell surface.

## m138 interacts with ICOSL

We then considered the possibility that m138 might be interacting with ICOSL, as it has been previously shown for CD80. To this end, we performed coimmunoprecipitation assays using NIH3T3 cells transiently expressing m138-GFP and HA-ICOSL. Proteins were pulled down by an anti-HA antibody and analyzed by western blot using an antibody against GFP. Taking into account the potential capacity of m138 to bind IgG, an immunoprecipitation was carried out with an anti-human Fc antibody as a control. As illustrated in *Figure 5D*, the m138 protein was recovered when the immunoprecipitation was carried out with the anti-HA antibody (lane 2), whereas when using the antibody against human Fc the viral protein was not detected (lane 1). As expected, after stripping the western blot and reprobing it with an anti-HA antibody, the 51 kDa specific ICOSL band was only observed in the precipitates obtained from the anti-HA mAb pull downs. These data suggest that m138, either directly or in an indirect manner, associates with ICOSL.

## Distinct m138 structural domains are required to downmodulate the two members of the B7 family, ICOSL and CD80

The ectodomain of m138 is predicted to be composed by three Ig-like domains (*Lenac et al., 2006*; *Corrales-Aguilar et al., 2014*). Previous work using mutant MCMVs containing m138 versions defective in either three potential Ig-like domains indicated different structural requirements in this viral protein for the downregulation of the NKG2D ligands (*Lenac et al., 2006*). We investigated if this was also the case for m138 to exert its effects on the two co-stimulatory molecules ICOSL and CD80, since this could allow us to dissect the impact of the viral protein on the two pathways of T-cell activation. Thus, we created three different m138-GFP mutants, missing the putative N-terminal Ig domain (Ig1; m138ΔIg1-GFP), the middle Ig domain (Ig2; m138ΔIg2-GFP), or both the middle and membrane-proximal (Ig3) Ig domains (m138ΔIg2/3-GFP) (*Figure 6A*, schematic representation of these constructs). Each m138-GFP mutant construct was transiently expressed on NIH3T3 HA-ICOSL cells and surface levels of ICOSL and CD80 were assessed by flow cytometry and compared to those present in cells transfected with the m138-GFP control. As shown in *Figure 6B*, the m138ΔIg1-GFP protein was incapable to significantly reduce ICOSL or CD80 cell surface levels. In contrast, while CD80 surface density decreased in cells transfected with m138ΔIg2-GFP and m138ΔIg2/3-GFP, these two m138 mutants did not alter in a substantial manner ICOSL expression. Therefore, we conclude that the Ig1 domain of m138 is absolutely required for both ICOSL and CD80 downregulation. However, while the putative Ig2 and Ig3 domains of m138 need to be preserved for the downregulation of ICOSL, they are dispensable for m138-mediated decrease of CD80 expression.

## m138 curtails ICOSL-dependent antigen presentation and subsequent CD8$^+$ T-cell responses during MCMV infection of APCs

The differential structural requirements of m138 to target CD80 and ICOSL permitted us to specifically address the functional consequences of the disruption of the ICOSL:ICOS pathway during the course of the MCMV infection. To this end, we employed two MCMV mutants containing m138 versions lacking either the Ig2 domain (MCMVm138ΔIg2), or the Ig2 and Ig3 domains (MCMVm138ΔIg2/Ig3). As it occurred in transfection assays with the m138 mutant plasmids defective in these Ig domains, infections with MCMVm138ΔIg2 or MCMVm138ΔIg2/Ig3 led to decreased cell-surface density of CD80 but not ICOSL (*Figure 7A*). We then performed an in vitro antigen presentation assay assessing the effect of these two MCMV mutants on the ability of APCs to induce an ICOSL-dependent CD8$^+$ T-cell response, in comparison to that of MCMVwt and MCMVΔm138. We co-cultivated BMDCs infected with the different viruses with naive CD8$^+$ T cells from splenocytes derived from T-cell receptor (TCR)-transgenic mouse line in which 90% of CD8$^+$ T-cells are specific for MCMV M38 peptide, and measured the percentage of IFN-γ$^+$ CD8$^+$ T cells 6 hr later (*Figure 7B*, schematic model) (Maxi mice; *Torti et al., 2011*). Remarkably, as illustrated in *Figure 7B*, both MCMVm138ΔIg2 and MCMVm138ΔIg2/Ig3-infected BMDCs stimulated significantly stronger IFN-γ responses by CD8$^+$ T cells as compared to the MCMVwt-infected cells. Moreover, the magnitude of

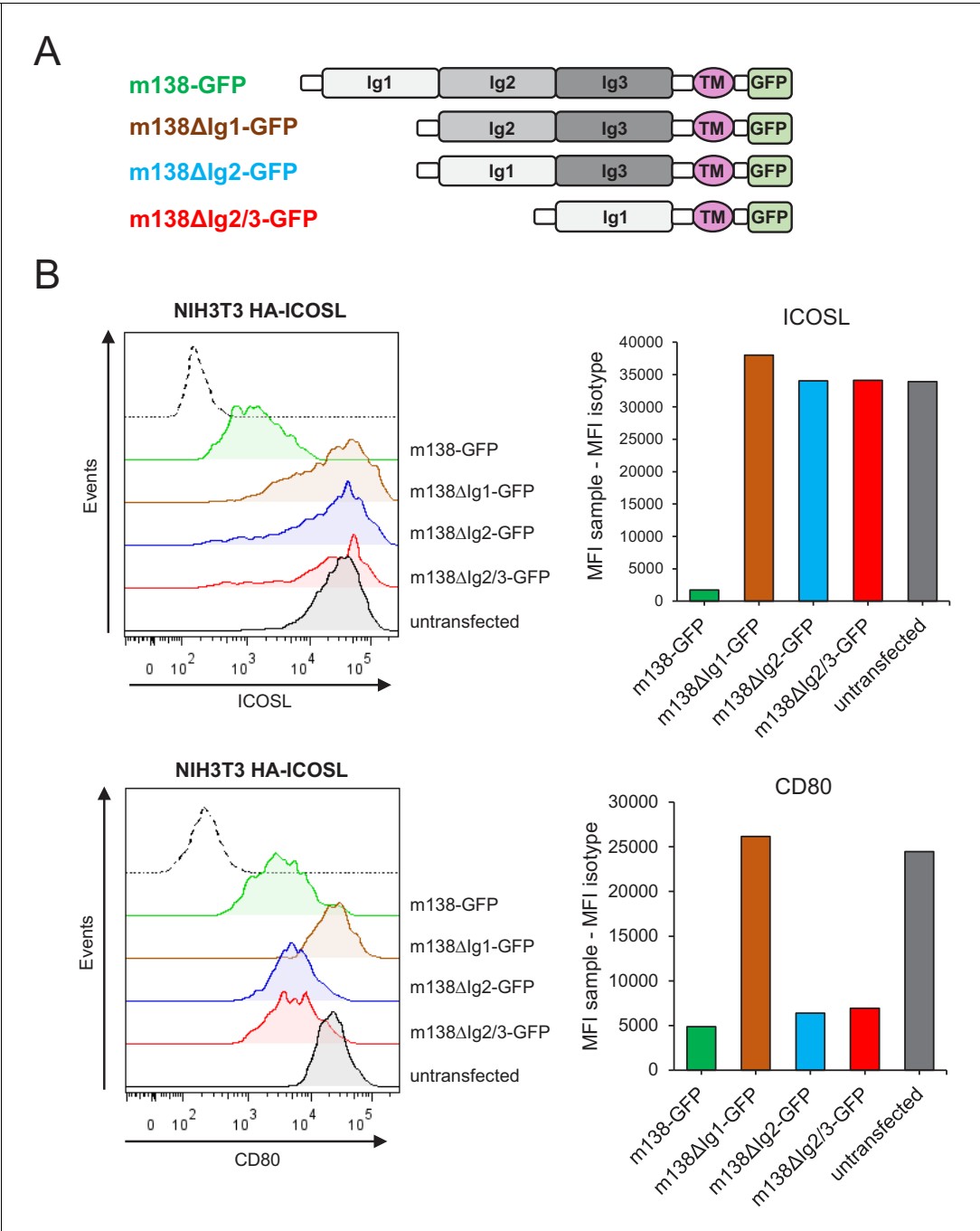

**Figure 6.** Identification of the m138 domains involved in the downmodulation of ICOSL and CD80. (**A**) Schematic representation (not drawn to scale) of m138-GFP protein mutants. (**B**) NIH3T3 ICOSL-HA cells were untransfected or transfected with m138-GFP, or m138-GFP mutant constructs and analyzed by flow cytometry for ICOSL and CD80 surface levels. Panels on the right represent the MFI of each sample minus the MFI of the control isotypes. Gray histograms represent the expression of untransfected cells and colored histograms represent the expression of cells transfected with m138-GFP (green), m138ΔIg1-GFP (brown), m138ΔIg2-GFP (blue), or m138ΔIg2/3-GFP (red). The isotype for each antibody was used as a negative control (dotted lines). A representative experiment out of two performed is shown.

the IFN-γ induction observed with these two MCMV mutants, which downregulated CD80 but preserved cell surface levels of ICOSL intact, almost approached the levels obtained in MCMVΔm138-infected BDMCs. These findings provide evidence for a substantial contribution of the m138 protein in the evasion of the CD8[+] T-cell immune response via manipulation of the ICOSL:ICOS axis (*Figure 7B*).

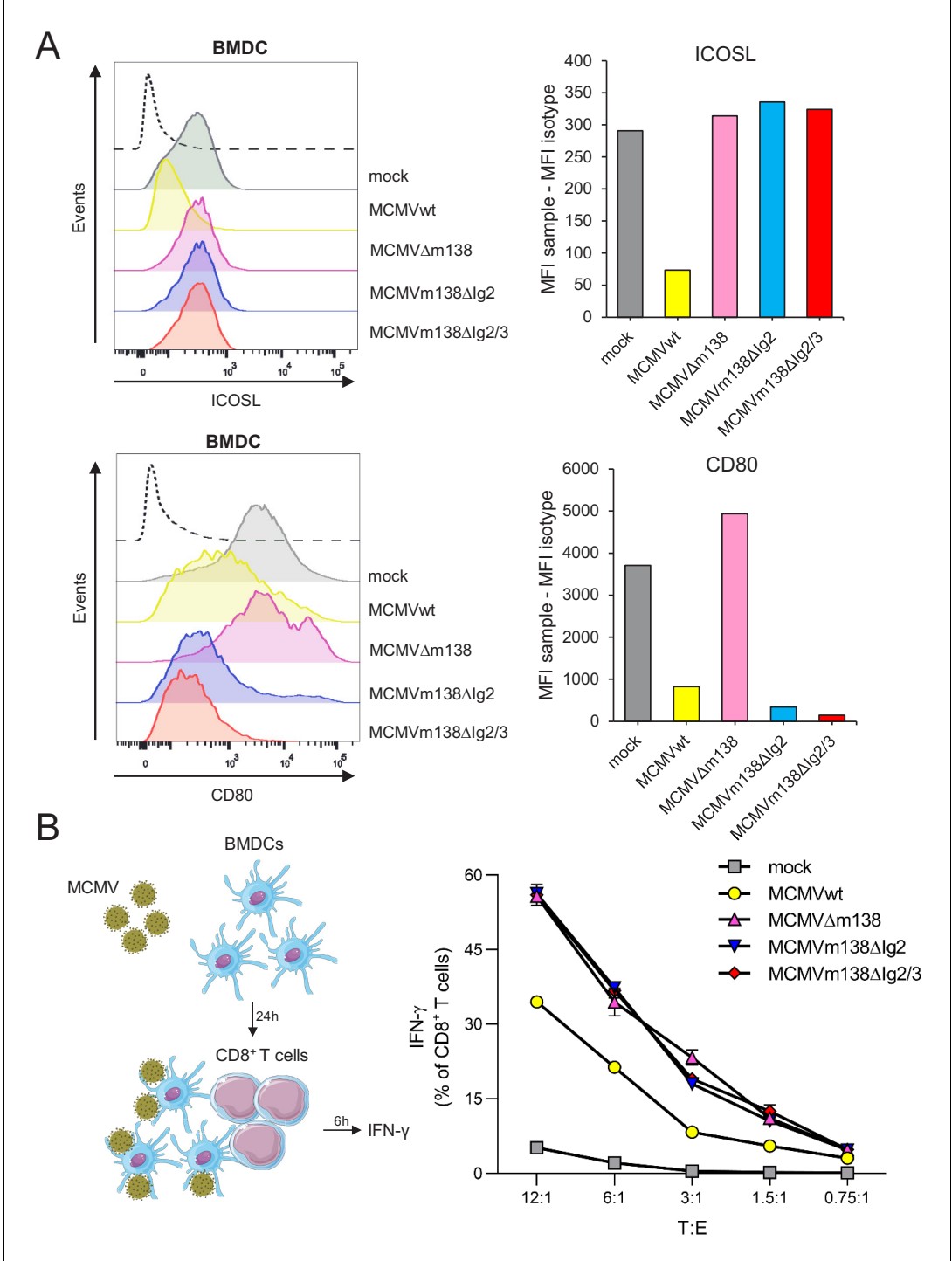

**Figure 7.** m138 is capable to restrict CD8[+] T-cell activation by downregulating ICOSL on the surface of antigen-presenting cells (APCs). (**A**) C57BL/6-derived BMDCs were mock-infected or infected for 24 hr with MCMVwt or MCMVm138 mutants at an moi of 3 and analyzed by flow cytometry for surface expression of ICOSL and CD80 using specific mAbs against each of these molecules. Gray histograms represent the expression of mock-infected cells and colored histograms represent the expression of cells infected (positive for the MCMV m04 protein) with MCMVwt (yellow), MCMVΔm138 (pink), MCMVm138ΔIg2 (blue), or MCMVm138ΔIg2/3 (red). The isotype for each antibody was used as a negative control (black line histograms). Panels on the right represent the MFI of each sample minus the MFI of the control isotypes. Results are representative of two independent experiments (**B**) On the left, schematic representation of the in vitro antigen-presentation assay. BMDCs infected as indicated in A with MCMVwt (yellow), MCMVΔm138 (pink), MCMVm138ΔIg2 (blue), or MCMVm138ΔIg2/3 (red), or left uninfected (gray) were further co-cultured with naïve CD8[+] T

*Figure 7 continued on next page*

*Figure 7 continued*

cells from Maxi mice. After 6 hr, IFN-γ production by CD8$^+$ T cells was determined by flow cytometry. Results are representative of three independent experiments. Results are expressed as the mean +/-SEM of the percentage values obtained for samples in each group.

## m138 impairs ICOS-dependent T-cell responses in vivo and limits virus control

We then explored the relevance of ICOSL interference by m138 during the acute MCMV infection. First, we sought to confirm that ICOSL downmodulation by MCMV was taking place in the in vivo context. Adult BALB/c mice were intraperitoneally (i.p.) infected with $2 \times 10^6$ PFU of MCMV-GFP, and 48 hr later the levels of ICOSL were analyzed in peritoneal macrophages by flow cytometry. As shown in *Figure 8A*, we observed a significant reduction of ICOSL levels, in parallel to those of MHC I used as a positive control, on the surface of the GFP+ MCMV-infected as compared to the uninfected GFP- macrophages. In addition, we directly assessed whether ICOSL was being specifically downregulated by m138 on peritoneal DCs during the in vivo infection. Thus, BALB/c mice were mock-infected or i.p. infected with $1 \times 10^6$ PFU of MCMVwt or MCMVΔm138, and at six hpi cell surface levels of ICOSL and CD80 were determined by flow cytometry on infected peritoneal DCs (positive for the MCMV protein m04) and compared to those on peritoneal DCs derived from mock-infected mice. *Figure 8B* illustrates a marked MCMVwt-induced downmodulation of ICOSL on peritoneal DCs, and a subsequent significant augmentation of the levels of this molecule in the absence of m138, on the surface of MCMVΔm138-infected peritoneal DCs. In the case of CD80, although the downmodulation caused by MCMVwt was subtle, a clear increase of CD80 levels was detected on MCMVΔm138-infected cells when compared to both mock-infected and MCMVwt-infected DCs. Therefore, these results confirmed a role of m138 downmodulating ICOSL and CD80 on primary mouse DCs (*Figure 8B*).

It has been previously described that the MCMV mutant depleted of m138 is attenuated early after infection in an NK-dependent manner, attributing this phenotype to the already reported activities of this viral protein downregulating NKG2D ligands (*Lenac et al., 2006*). Taking in consideration m138 effects on the two co-stimulatory molecules CD80 and ICOSL, we sought to assess the contribution of m138 at later times of the acute infection by comparing the in vivo phenotype of MCMVm138ΔIg2 with that of MCMVΔm138 and MCMVwt under conditions that included depletion of NK cells or combined depletion of NK cells and CD4$^+$ and CD8$^+$ T cells. Of note, previous work with MCMVm138ΔIg2 showed that this mutant virus was capable to downregulate NKG2D ligands MULT-1 and RAE-1ε, but not H60 cell surface molecules (*Arapović et al., 2009*; *Lenac et al., 2006*). The information of the effects that MCMVwt and mutants have on these m138 targeted host molecules on the surface of infected cells is indicated in part C of *Figure 8*. C57BL/6 mice were intravenously (i.v.) infected with $2 \times 10^5$ PFU of the different viruses, left untreated or treated with the antibodies to deplete specific lymphocyte subsets, and viral titers in salivary glands and lungs were quantified 14 days post-infection (dpi) (*Figure 8D*, schematic representation of the assay). As illustrated in *Figure 8D*, in the salivary glands, viral titers of MCMVm138ΔIg2, a mutant which is unable to downregulate H60 and ICOSL, were lower than those of MCMVwt. In fact, the MCMVm138ΔIg2 titers in this organ were below the detection limits for all animals, in a similar way as in MCMVΔm138 infected mice. NK cell depletion led to a substantial increase of both MCMVΔm138 and MCMVm138ΔIg2 infection in this organ, but importantly, MCMVm138ΔIg2, which was downmodulating CD80 but not ICOSL, was still heavily attenuated as compared to MCMVwt under these conditions. This finding indicated that ICOSL targeting by m138 is important for the virus to increase its load in salivary glands. Conversely, the downregulation of CD80 by m138 did not seem to play a major role during MCMV replication in this organ. Furthermore, when besides NK cells, CD4$^+$ and CD8$^+$ T cells were depleted, MCMVm138ΔIg2 growth in the salivary glands was completely restored, indicating that m138 via interference with ICOSL evades T-cell responses. It is important to note that when NK cells together with CD4$^+$ and CD8$^+$ T cells were depleted, MCMVΔm138 seemed to be slightly attenuated. This may suggest that m138 is capable to perform additional immune evasion activities independent of NK and T cells. Similar results were obtained when viral titers were evaluated in the lungs of the infected animals, although in this case the contribution of m138 via ICOSL in the control of the MCMV infection seemed to be less significant (*Figure 8D*).

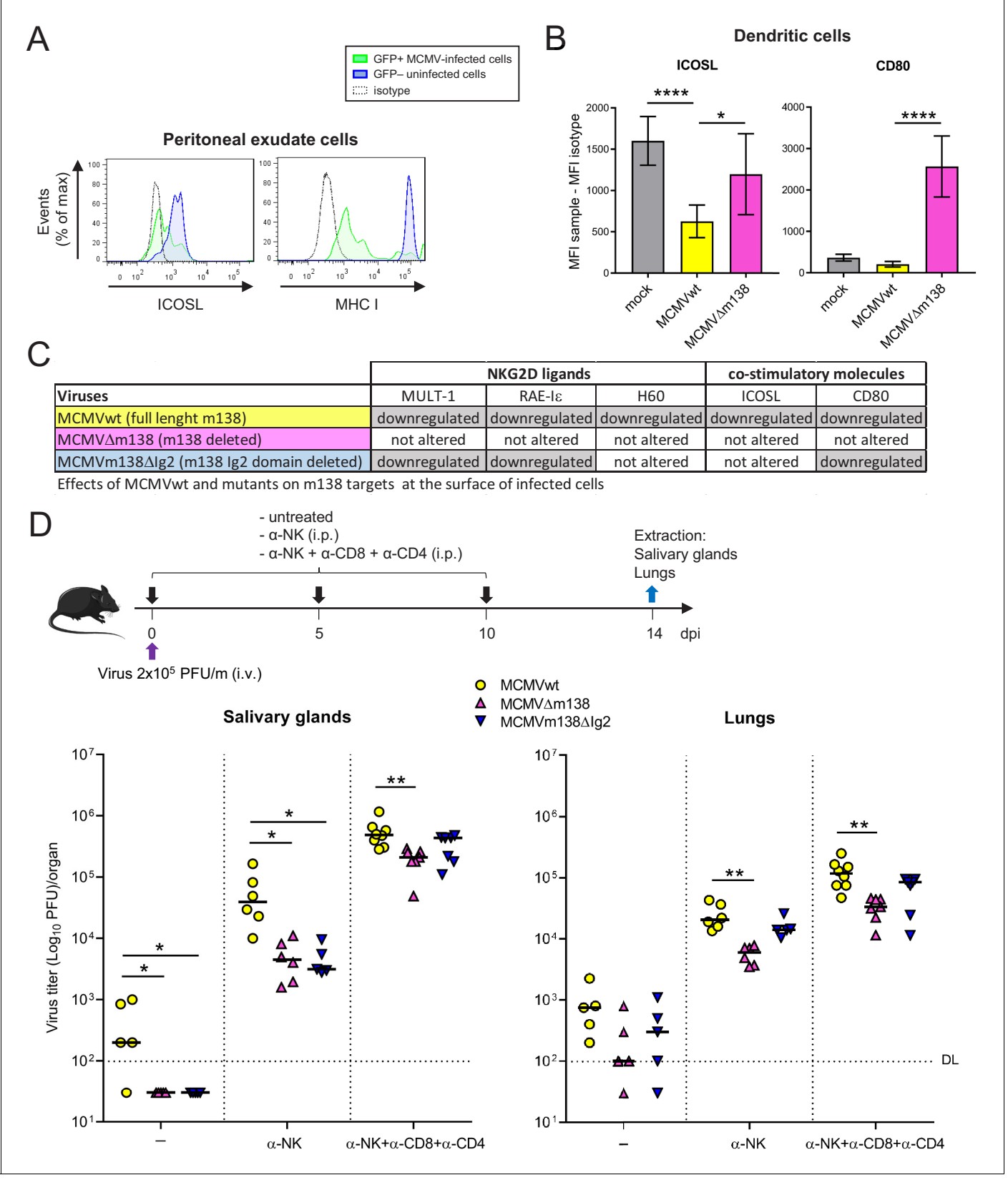

**Figure 8.** Via targeting ICOSL, m138 reduces T-cell mediated control of MCMV infection in vivo. (**A**) BALB/c mice were intraperitoneally (i.p.) inoculated with 2 × 10⁶ PFU MCMV-GFP. Two days post-infection mice were sacrificed and peritoneal exudate cells were extracted and analyzed by flow cytometry for surface expression of ICOSL and MHCI, using specific mAbs against each of these molecules. Blue histograms represent the expression

*Figure 8 continued on next page*

*Figure 8 continued*

of uninfected (GFP-) cells and green histograms represent the expression of MCMV-infected (GFP+) cells from the same mouse. The isotype for each antibody was used as a negative control (dotted lines). The results obtained from a representative infected mouse out of two are shown. (B) BALB/c mice (n = 4/group) were mock-infected or i.p. infected with $1 \times 10^6$ PFU of MCMVwt or MCMV$\Delta$m138. At 6 hpi, peritoneal exudate cells were extracted and surface expression of ICOSL and CD80 assessed by flow cytometry on the surface of DCs (CD11+ MHC II+ CD3-CD19- NKp46- cells), gating on m04-positive cells when derived from MCMVwt- or MCMV$\Delta$m138-infected mice. Results are expressed as the mean +/-SD of the MFI values obtained for samples from two independent experiments. (C) Table displaying the immunomodulatory effects of the different MCMVs on the cellular targets of m138. (D) On the top, a schematic representation of the MCMV$\Delta$m138 and MCMVm138$\Delta$Ig2 in vivo infection assay is shown. C57BL/6 mice (n = 5–7/ group) with or without NK, or NK, CD4+ and CD8+ T-cell depletion, as indicated, were intravenously (i.v.) inoculated with $2 \times 10^5$ PFU/mouse of MCMVwt (yellow circles), MCMV$\Delta$m138 (pink triangles) or MCMVm138$\Delta$Ig2 (blue triangles). At day 14 post-infection mice were sacrificed and viral titers in salivary glands and lungs of individual mice were determined by standard plaque assays. Horizontal bars indicate the median values. The Kruskal-Wallis test was used to assess statistical differences between experimental groups. *p<0.05, **p<0.01. A representative experiment out of two performed is shown.

The online version of this article includes the following figure supplement(s) for figure 8:

**Figure supplement 1.** m138 contributes to immune evasion of CD4+ T cells in spleens and lungs of MCMV-infected mice.

MCMV has been reported to induce the accumulation of virus-specific IL10+ CD4+ T cells in an ICOSL-dependent manner in salivary glands (*Clement et al., 2016*). Thus, we directly addressed the impact of m138 on the evasion of MCMV-specific CD4+ T cells responses in various organs. To this end, we adoptively transferred TCR-transgenic CD4+ T cells specific for the M25 protein (M25-II; *Mandaric et al., 2012*) into C57BL/6 SCID mice, depleted them of NK cells, and 1 day later infected them with $2 \times 10^5$ PFU of MCMVwt or MCMV$\Delta$m138 (See schematic model in *Figure 8—figure supplement 1*). Seven days after infection, the expansion of CD4+ T cells was determined in the spleen, lungs, and salivary glands. The data, illustrated in *Figure 8—figure supplement 1*, show that the frequencies of CD4+ T cells increased in the three organs analyzed upon MCMV infection. Moreover, a role of the viral m138 protein participating in immune evasion of CD4+ T cells could be observed in spleen and lungs when comparing data from MCMVwt and MCMV$\Delta$m138 infections, although this was not the case in the salivary glands.

## The ICOSL:ICOS axis contributes to the generation of MCMV-specific humoral immune responses in vivo

We also sought to determine the importance of ICOSL-mediated co-stimulation for mounting MCMV-specific T-dependent antibody responses. We approached this, by inhibiting ICOSL:ICOS interaction during the acute MCMV$\Delta$m138 infection via the administration of a blocking anti-ICOSL antibody. Groups of BALB/c mice were infected with $2 \times 10^6$ PFU of MCMV$\Delta$m138 and treated with or without an anti-ICOSL mAb, starting 1 day prior infection. Salivary gland draining lymph nodes and spleens were extracted in order to analyze by flow cytometry different T and B lymphocyte subsets, and blood collected, at day 14 post-infection. As illustrated in *Figure 9A*, a marked decrease of the percentage of Tfh cells in both spleen and lymph nodes was observed when infected mice were deprived of ICOSL-mediated signals. In addition, while the percentage of follicular B cells was not substantially altered, or was even slightly higher, upon ICOSL blockade, GC B cells in lymph nodes were dramatically reduced. Consistent with these observations, an analysis by immunofluorescence of splenic sections revealed that GCs were smaller in size in the infected mice that received anti-ICOSL antibody as compared to mice that did not received it, whereas this was not the case of the follicles, whose sizes seemed not to be affected by the treatment (*Figure 9B*). These results, which are in agreement with the phenotype associated with ICOS absence, pointed to a disturbed production of MCMV-specific antibodies during the primary infection under conditions of ICOSL blockade. No significant differences in the percentage of other cell subsets such as marginal zone B cells, B1 B cell subsets, plasma cells, or overall CD4+ and CD8+ T cells between the two groups of infected animals were detected (*Figure 9—figure supplement 1*). In addition, the blockade of ICOSL also resulted in an increased percentage of naïve CD4+ T cells in the spleen and decreased percentages of memory CD4+ T cells in spleens and lymph nodes (*Figure 9A*), an indication that long lasting immune protection to MCMV could be also impaired in the absence of ICOSL:ICOS signaling.

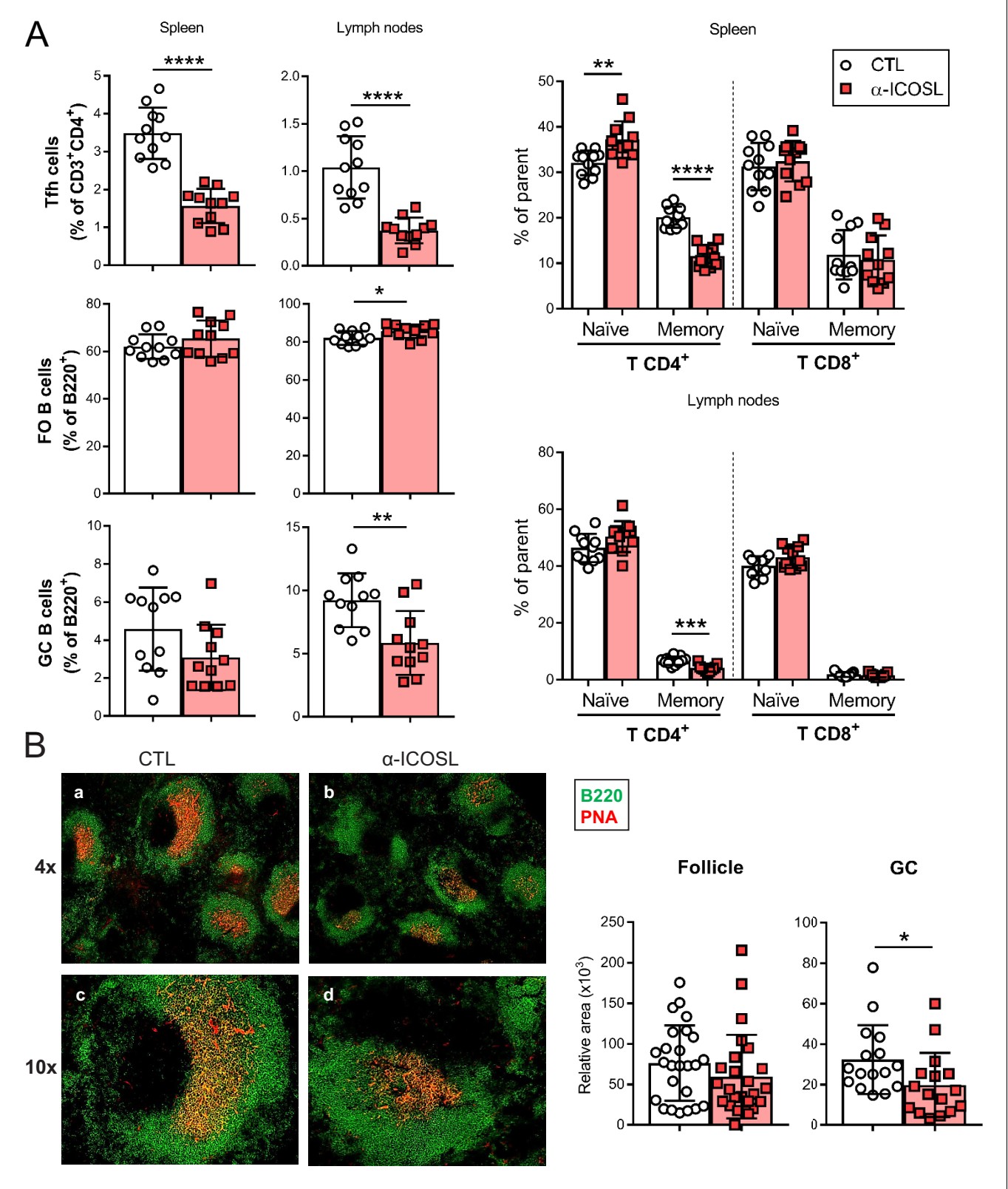

**Figure 9.** Blockade of ICOSL reduces the frequency of particular lymphocyte subsets in the spleen and lymph nodes of MCMV-infected mice. (**A**) BALB/c mice (n = 11–12/group) were i.p. inoculated with 2 × 10⁶ PFU of MCMVΔm138 and treated with (α-ICOSL) or without (CTL) 100 mg of anti-ICOSL mAb. At day 14 post-infection mice were sacrificed, and spleens and lymph nodes were isolated and disaggregated. Cell suspensions were analyzed by

*Figure 9 continued on next page*

Figure 9 continued

flow cytometry using anti-mouse CD3, CD4, CD8, CD44, CD62L, CXCR5, and PD1 for T-cell phenotype. For B cell phenotype B220, CD21, CD23, CD95, and GL7 mAbs were used. Percentages of Tfh cells (CD3$^+$CD4$^+$CXCR5$^{hi}$PD1$^{hi}$), germinal center (GC) B cells (CD95$^{hi}$GL7$^{hi}$B220$^+$), follicular (FO) B cells (CD21$^+$CD23$^+$B220$^+$), and CD3$^+$CD4$^+$ or CD3$^+$CD8$^+$ naive (CD44$^{low}$CD62L$^{hi}$) and memory (CD44$^{hi}$CD62L$^{low}$) T cells from these two organs are shown. (B) Representative immunofluorescence of spleens from mice infected and treated with (panels b and d) or without (panels a and c) the anti-ICOSL mAb, as indicated in A. Two color colocalization with B220 in green and peanut agglutinin (PNA) in red are shown at ×4 and ×10 magnifications. Panels on the right show a graphic representation of follicle and GC relative areas from tissue sections of a representative mouse of each experimental group. In graphs shown in A and B, results are expressed as mean +/-SD. Mice treated with anti-ICOSL mAb are represented as red squares and mice not receiving the mAb are represented as open circles. Two-tailed unpaired *t*-tests were used to assess statistical differences between experimental groups. *p<0.05, **p<0.01, ***p<0.001, ****p<0.0001. Data are pooled from two independent experiments.

The online version of this article includes the following figure supplement(s) for figure 9:

**Figure supplement 1.** Effect of ICOSL blockade on the frequency of several lymphocyte subsets in the spleen and lymph nodes of MCMV-infected mice.

Due to the above results and the implication of ICOS in promoting T-dependent antibody responses, we assessed by ELISA whether the levels of MCMV-specific antibodies in the sera of the infected mice were altered after ICOSL blockade. We observed that mice that received the anti-ICOSL antibody had significantly reduced levels of total IgG specific for MCMV than those that did not receive it (*Figure 10A*). When we evaluated the serum levels of the predominant subclasses of natural antibodies, a substantial decreased production of MCMV-specific IgG1 and IgG2a and b was detected in infected animals in which ICOSL was blocked, while no differences were found in the production of MCMV-specific IgG3 or IgM responses between both groups (*Figure 10A*). These results indicated that ICOSL-mediated co-stimulation was crucial for the development of MCMV-specific IgG1 and IgG2 responses.

The observed alterations in the humoral responses prompted us to conduct in vitro neutralization assays of MCMV with the serum of the infected mice. As illustrated in *Figure 10B*, sera from mice that received the anti-ICOSL antibody exhibited a significant lower capacity to neutralize MCMV compared to those that did not received it. Similar findings were obtained when the neutralization assays were performed in the presence of complement (*Figure 10—figure supplement 1*). In conclusion, our findings indicate a prominent role of ICOSL:ICOS interactions in the induction of effective T and B cell responses during MCMV infection.

## HCMV reduces cell surface levels of ICOSL on APCs

We then investigated if HCMV has also the capacity to interfere with cell surface expression of ICOSL in APCs. To this end, human primary monocyte-derived macrophages were infected with HCMV-GFP (TB40/E strain), and analyzed by flow cytometry 72 hr after infection. In addition, we tested the viral effects on the THP-1 cell line differentiated by treatment with the protein kinase-C agonist phorbol-12-myristate-13-acetate (PMA). As shown in *Figure 11A*, in both cell types, HCMV infection resulted in a marked reduction in cell surface ICOSL expression compared to non-infected cells in the same culture or to mock-infected cells. In contrast, cell surface expression of CD70 remained unaffected, indicating that ICOSL molecules were specifically targeted in HCMV-infected cells. Kinetic assays performed in THP-1 cells indicated that by 24 hpi ICOSL levels were already drastically reduced (*Figure 11B*). As shown in *Figure 11C*, the fact that upon infection with UV-inactivated HCMV ICOSL density at the plasma membrane was not significantly different from that of mock-infected control cells, suggested that the expression of one or more viral gene products were required to downmodulate ICOSL. We also analyzed whether the total levels of ICOSL were altered upon infection, by carrying out immunoblot analysis on uninfected or HCMV-infected protein lysates using an anti-ICOSL polyclonal antibody. As can be seen in *Figure 11D*, ICOSL in the THP-1 uninfected cells migrated as a broad band of approximately 70 kDa. In contrast, after infection with HCMV, the levels of ICOSL were drastically reduced. Interestingly, we observed that treatment with leupeptin and bafilomycin was not able to significantly restore ICOSL expression.

## Additional herpesviruses also target cell surface ICOSL

Finally, we asked whether, similar to murine and human CMVs, other human herpesviruses are also able to alter ICOSL cell-surface expression during the course of the infection. We focused on HSV-1

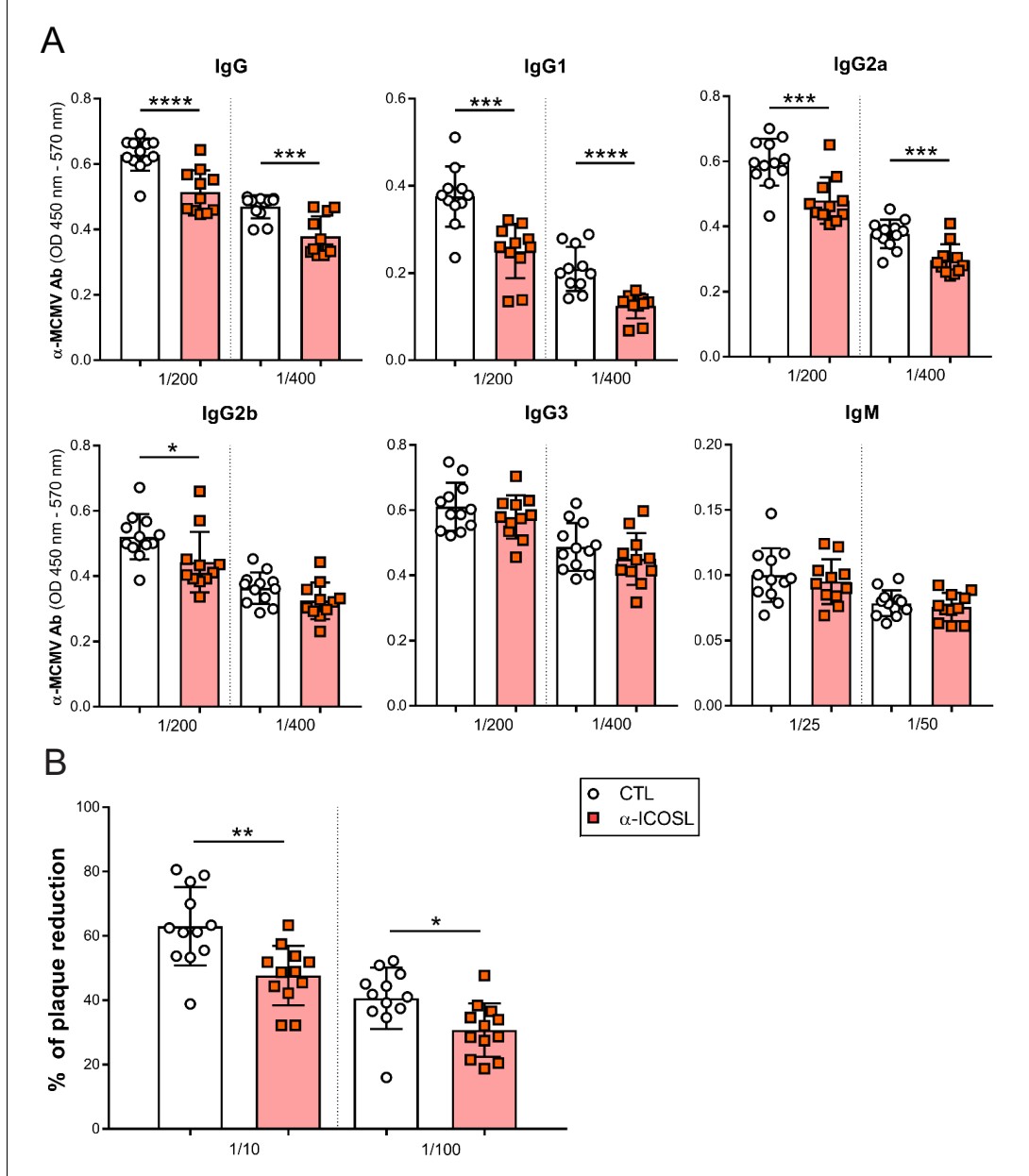

**Figure 10.** ICOSL blockade limits the generation of MCMV specific antibodies. (**A**) Total levels of IgG, IgG1, IgG2a, IgG2b, IgG3 (at dilutions 1:200 and 1:400), and IgM (at dilutions 1:25 and 1:50) were determined by ELISA from sera collected at day 14 of the same infected BALB/c mice indicated in *Figure 8A*. (**B**) Sera (at dilutions 1:10 and 1:100) as in A were tested in a viral neutralization assay. Results are presented as the percentage of plaque reduction determined by the ratio of the number of plaques counted in the sample wells relative to the ratio of plaques in wells containing the serum of an uninfected mouse used as a negative control. In A and B, results are expressed as mean +/- SD. Mice treated with anti-ICOSL mAb are represented as red squares and mice that did not receive the mAb are represented as open circles. Two-tailed unpaired *t*-tests were used to assess statistical differences between experimental groups. *p<0.05, **p<0.01, ***p<0.001. Data are pooled from two independent experiments.

The online version of this article includes the following figure supplement(s) for figure 10:

**Figure supplement 1.** Effect of ICOSL blockade on MCMV neutralization in the presence of complement.

and HSV-2, and infected THP-1 cells using GFP versions of these viruses. Notably, as illustrated in *Figure 11E*, the levels of ICOSL were reduced during infection of both HSV-1-GFP and HSV-2-GFP compared to those of mock-infected cells. In contrast, surface expression of CD70 did not change, confirming the specificity of the findings. Moreover, the observation that upon infection of THP-1

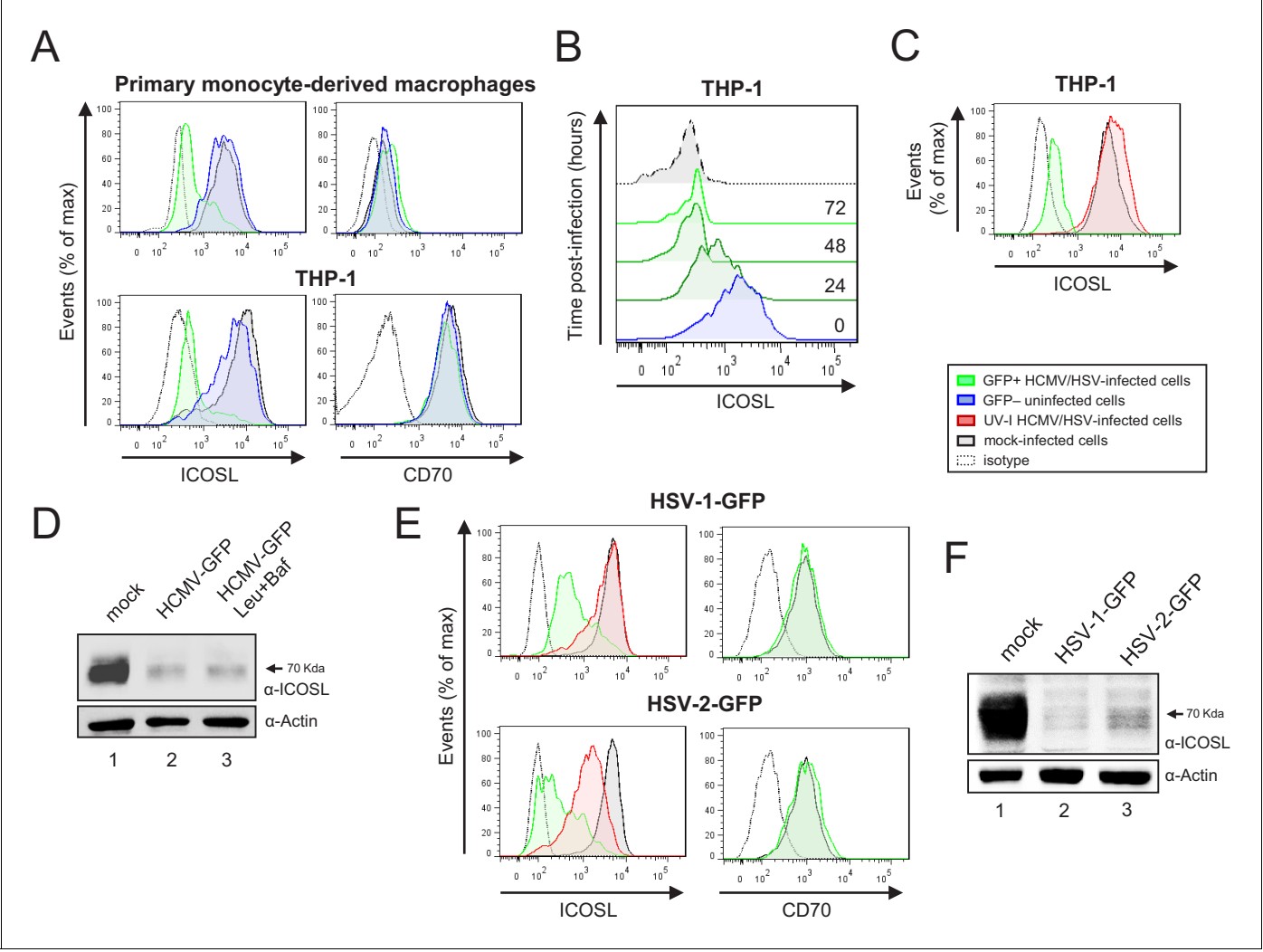

**Figure 11.** HCMV, HSV-1, and HSV-2 also limit cell surface expression of ICOSL on antigen-presenting cells (APCs). (A) Primary monocyte-derived macrophages (upper panels) and PMA-treated THP-1 cells (bottom panels) were mock-infected or infected for 72 hr with HCMV-GFP at an moi of 10 and analyzed by flow cytometry for cell-surface expression of human ICOSL or CD70 using specific mAbs against each of these receptors. Gray histograms represent the expression of mock-infected cells, green histograms represent the expression on HCMV-infected (GFP+) cells, and blue histograms represent the expression on uninfected (GFP-) cells from the same culture. (B) PMA-treated THP-1 cells were mock-infected (time 0) or infected with HCMV-GFP as in A and analyzed by flow cytometry for surface expression of ICOSL at the different time points after infection indicated. (C) Same as in A, except that an moi of 20 was used, and THP-1 cells were also exposed for 72 hr to the same amount of HCMV-GFP UV-inactivated (red histogram). (D) Equal amounts of lysates from PMA-treated THP-1 cells mock-infected (lane 1) or infected for 72 hr at an moi of 20 with HCMV-GFP (lanes 2 and 3), and when indicated, treated with 250 μM leupeptin and 20 nM of bafilomycin A1 (lane 3), were lysed and analyzed by western blot with antibodies against ICOSL and actin, followed by anti-rabbit IgG-HRP (ICOSL) or anti-mouse IgG-HRP (actin). (E) PMA-treated THP-1 cells were mock-infected or infected with HSV-1-GFP, HSV-1-GFP UV-inactivated, HSV-2-GFP or HSV-2-GFP UV-inactivated at an moi of 100 (HSV-1) or 200 (HSV-2). Twenty-four hr later, the expression of human ICOSL and CD70 were analyzed as in A. Green histograms represent the expression of ICOSL on infected cells, red histograms represent the expression on UV-inactivated HSV infected cells, and gray histograms represent the expression on mock-infected cells. The isotype for the ICOSL antibody was used as a negative control (dotted lines). A representative experiment out of three performed is shown. (F) PMA-treated THP-1 cells were mock-infected (lane 1) or infected with HSV-1-GFP (lane 2) and HSV-2-GFP (lane 3) as indicated in E. Cell lysates were prepared, and subjected to western blot analysis using anti-human ICOSL or anti-actin mAbs as in D. The molecular weight of the band corresponding to ICOSL is indicated in D and E in kilodaltons on the right margin.

cells with UV-inactivated HSV-1 and HSV-2, only a partial downregulation of ICOSL was observed, indicated that these two human viruses also harbor genes targeting ICOSL (*Figure 11E*). Last, when we evaluated by western blot ICOSL expression in whole THP-1 cell lysates, a clear reduction of the levels of this molecule could be appreciated upon infection with both HSV-1 and HSV-2

(*Figure 11F*). Altogether, these findings demonstrate that ICOSL downregulation is a strategy used by different herpesviruses.

## Discussion

Co-stimulatory molecules play a decisive role in shaping the extent and nature of the cell-mediated adaptive immune response by tuning T-cell activation. Therefore, it is not surprising that pathogens, including viruses, have devised mechanisms to abrogate their signaling and blunt host responses that would otherwise limit their replication (*Khan et al., 2012*). In this study, we demonstrate that the ligand for the co-stimulatory molecule ICOS is dramatically diminished from the surface of APCs during infection by different herpesviruses. With a primary focus on MCMV, the most widely used model to study CMV infection, we show that the downregulation of ICOSL restricts the magnitude of MCMV-specific T-cell responses and potentiates viral burden in vivo.

We describe here that ICOSL expression is slightly induced upon cell exposure to MCMV, presumably due to viral sensing by pattern recognition receptors. However, MCMV rapidly removes this molecule from the surface of infected cells by expressing m138. Infections with an MCMV lacking the *m138* gene, and assays with the viral gene expressed in isolation, indicate that m138 is necessary and sufficient to decrease ICOSL cell surface levels. The early expressed m138 protein, which is an Fc receptor homologue, has been shown not only to bind the constant Fc domain of IgG, but also to downmodulate the expression of three cellular ligands of the activating NKG2D receptor, RAE-1ε, H60, and MULT-1, and another B7 family molecule, CD80 (*Arapović et al., 2009*; *Lenac et al., 2006*; *Mintern et al., 2006*; *Thäle et al., 1994*). Consequently, this viral protein holds the potential to control the antiviral function of NK and T cells, as well as the humoral response. Hence, the m138 early glycoprotein provides an excellent example of how CMVs have refined proteins to execute multiple immune-evasion functions. It is becoming increasingly clear that the evolution of multifunctional proteins is not only a hallmark of RNA viruses, with limited genome sizes and relatively small number of genes, but that it is also employed by large DNA viruses to make optimal use of their coding capacity. For instance, MCMV produces the multifaceted immunomodulatory protein m152, which is capable of downregulating MHC class I molecules and different RAE-1 isoforms, as well as modulating the cGAS-STING pathway, thereby evading type I IFN-, NK-, and T cell-dependent immune responses to MCMV infection (*Fink et al., 2013*; *Krmpotic et al., 1999*; *Lodoen et al., 2003*; *Stempel et al., 2019*; *Ziegler et al., 1997*). Moreover, we have recently demonstrated that by interfering with AP-1-mediated protein sorting, the m154 glycoprotein targets a broad-spectrum of cell surface molecules implicated in the antiviral NK and T-cell responses (*Strazic Geljic et al., 2020*; *Zarama et al., 2014*). In HCMV, this concept is best exemplified by the *US12* family, whose members, as it will be discussed below, have been reported to alter the expression of numerous plasma membrane proteins, mainly NK ligands, adhesion proteins and cytokine receptors (*Fielding et al., 2017*). m138 is a 69 kDa type I transmembrane glycoprotein, largely localized in the ER and lysosomal compartments, and shown to be further processed into a 105 kDa highly glycosylated form (*Mintern et al., 2006*).

Based on the ability of MCMV-infected cells to bind IgG, m138 was reported to be a cell surface resident protein, a feature shared by the different viral Fcγ receptors (*Corrales-Aguilar et al., 2014*; *Lenac et al., 2006*). Consistent with its location at the plasma membrane, the viral protein was shown to perturb the endocytosis of surface RAE-1ε and MULT-1, interfering with the clathrin dependent endocytosis of this later cellular target, altering its recycling and leading to its subsequent degradation in lysosomes (*Arapović et al., 2009*; *Lenac et al., 2006*). A different mode of action was reported for m138 in the downregulation of CD80, targeting the cellular molecule when newly synthesized early in the secretory pathway and mislocalizing it to lysosomal compartments (*Mintern et al., 2006*). To date, studies on the maturation and posttranslational modifications of ICOSL are still lacking, but our observations are compatible with the notion that, as in the case of CD80, m138 interacts with ICOSL, preventing it to mature and reach the plasma membrane, and driving this molecule to lysosomal degradation. Accordingly, we found that during MCMV infection, m138 and ICOSL colocalize in intracellular compartments, where the viral protein is primarily expressed, and that upon treatment with lysosomal inhibitors the levels of both ICOSL and m138 augment. Moreover, our co-precipitation experiments demonstrate that indeed m138 directly associates with ICOSL. The details on how the interactions of m138 with its structurally diverse targets

can potentially occur remain to be elucidated. In this regard, solving the crystal structure of m138 alone or bound to its cellular targets may be of great value for understanding its versatility and different modes of action.

Due to the multiple cellular targets of m138, dissecting the functional consequences of lessening ICOSL cell surface levels on virally infected cells appeared to be complicated. m138 was predicted to contain an ectodomain with three putative Ig-like domains, exhibiting a relatively low but yet significant sequence homology with the Ig-like domains of murine FcγRs CD16 and CD32 (*Budt et al., 2004*; *Lenac et al., 2006*). Previous work indicated distinct m138 structural requirements to down-modulate the different NKG2D ligands (*Arapović et al., 2009*; *Lenac et al., 2006*; *Mintern et al., 2006*). Thus, by generating versions of the m138 protein devoid of some of its Ig domains, and through the use of MCMV recombinants bearing similar deletions within the viral protein, we determined that the downregulation of the two B7-molecules, ICOSL and CD80, was dependent on the N-terminal m138 Ig domain. However, while the middle and membrane-proximal Ig domains of m138 were largely dispensable for disturbing CD80, they needed to be preserved for ICOSL cell surface removal. This result indicates a distinct mode of action of m138 to target the two costimulatory molecules, and gave us the opportunity to separate the corresponding impacts on T-cell activation. Optimal T-cell activation is characterized by rapid proliferation, cytokine production, and efficient effector functions. Importantly, employing an in vitro antigen presentation assay and the m138 mutant MCMVs, we showed that via ICOSL downmodulation, m138 contributes to impair antiviral CD8$^+$ T-cell responses. This agrees with the earlier evidence that the m138 protein expressed in isolation is capable to reduce the ability of DCs to promote T-cell activation in a CD80-dependent manner, and the fact that applying a blocking anti-CD80 antibody only partially reverted these effects (*Mintern et al., 2006*). Our observations, however, in the context of the infection, point to a prominent role of the ICOSL-dependent m138 function compromising T-cell stimulation.

The m138 protein has not an ortholog in HCMV. However, we demonstrate that following a productive HCMV infection of primary monocyte-derived macrophages or differentiated THP-1 cells, which express high levels of ICOSL, a strong ICOSL cell surface downregulation also takes place, and that this process requires viral gene expression. Using a multiplexed proteomic approach and HCMVs with deletions of each individual ORF of the *US12* viral family, Fielding and co-workers had previously described that members of this family selectively target a number of plasma membrane proteins crucial for NK cell activity, adhesion, and cytokine signaling during HCMV infection of human fibroblasts (*Fielding et al., 2017*). Interestingly, in these assays, and despite the low levels of ICOSL in human fibroblasts, no reduction of cell surface ICOSL could be observed after infection with HCMVs deleted in either US16 or US20, strongly suggesting that these two HCMV products are involved in ICOSL downregulation. Our findings also indicate that HCMV not only leads to diminished cell surface ICOSL, but also to an abrogation of its expression in whole cell levels, via a lysosomal degradation-independent process. Importantly, a similar complete loss of ICOSL on differentiated THP-1 cells was observed during infection by two α-herpesviruses, HSV-1 and HSV-2. In this case, though, both the mechanism as well as the identity of the viral proteins involved in this process remain to be defined.

The importance of the ICOSL:ICOS pathway has been explored in several models of bacterial and parasitic infections (*Wikenheiser and Stumhofer, 2016*). However, to date, only a limited number of studies on murine models of infection with RNA viruses (LCMV, VSV, and influenza virus), in conditions of ICOS deficiency or blockade with an ICOSL-Ig molecule, have been reported, and shown that ICOS triggering plays a marginal although significant role in the development of CD4$^+$ T-cell effector responses (*Bertram et al., 2002*; *Kopf et al., 2000*). But, persisting DNA viruses, with infections more dependent on cell-mediated immune control, may differ. Besides the role played by ICOS in directing effector T-cell activation, proliferation, and differentiation, this molecule has been crucially implicated in the generation of T-dependent antibody responses. Thus, in this study, we examined whether anti-MCMV humoral immune responses were altered in the absence of ICOS signaling. Overall, our data employing a blocking monoclonal anti-ICOSL antibody in the context of the acute MCMV infection revealed a high importance of the ICOSL:ICOS axis in the formation of Tfh cells and subsequent GC B-cell development in spleens and lymph nodes follicles, consistent with what has been previously observed in other in vivo experimental models of infection (*Wikenheiser and Stumhofer, 2016*). Importantly, as a consequence of the observed effects, ICOSL blockade was associated with a lower production of MCMV-specific IgG1 and IgG2 antibodies.

Furthermore, a decreased capacity of the antibodies generated to neutralize MCMV was also observed. Notably, while the CD80/CD86:CD28 co-stimulatory pathway has been shown to be also involved in the induction of Tfh cells and class switching of MCMV-specific antibodies, we find that this pathway is not capable to compensate the defects observed in the absence of ICOS signaling, evidencing the non-redundant role of the two axis in this scenario (*Welten et al., 2016*). We have not directly addressed here whether the lack of ICOSL co-stimulation is associated with a deficient viral control, due to the highly attenuated phenotype of the MCMV defective in m138 employed in the study. Remarkably, a reduction of splenic CD4⁺ memory T cells was also observed in our assays upon ICOSL blockade, further supporting the potential significance of the ICOSL:ICOS axis in the long-term protection against MCMV. Of interest, ICOS- and ICOSL-deficient patients have been associated with combined T- and B-cell immunodeficiency, and importantly, with an increased susceptibility to viral infections (*Roussel et al., 2018*; *Schepp et al., 2017*). Moreover, while the infectious phenotype of these patients varied, a high incidence of recurrent herpesvirus infections, mostly including herpes simplex and cytomegalovirus infections, was observed.

It was previously shown that deletion of m138 or both m138 and the CD86 downmodulator m147.5 from MCMV has a dramatic impact on viral load during acute infection (*Arens et al., 2011*; *Crnković-Mertens et al., 1998*). To directly address the potential importance of m138-mediated downregulation of ICOSL on viral growth in vivo, we employed the recombinant MCMV expressing m138 without the middle Ig domain, and therefore able to downmodulate CD80 and CD86 but not ICOSL in infected APCs. The fact that under conditions of NK cell depletion the viral mutant was still attenuated at late times post-infection, particularly in the salivary glands, and the nearly complete reversion of this defective phenotype in the absence of NK cells and T cells, indicated that m138 by interfering with ICOSL contributes to subvert T-cell responses in vivo. By performing adoptive transfer assays of exogenous M25-specific transgenic CD4⁺ T cells into C57BL/6 SCID mice, which were depleted of NK cells, and infected with MCMVwt or the MCMV lacking m138, we evaluated the impact that the viral protein may have on immune evasion of CD4⁺ T cells in various organs. We observed that m138 clearly contributes to diminish M25-specific CD4⁺ T cell proliferation in spleen and lungs of infected mice, but not in salivary glands, where virus specific CD4⁺ T cells are important for control of the viral infection. It must be noted, however, that since in the study we only measured M25-specific responses, we cannot discard that in the salivary glands m138 may be influencing in a different way CD4⁺ T cell responses specific for other MCMV epitopes. Altogether, our results support the notion that downmodulation of ICOSL by m138 represents an effective immune evasion strategy during the acute MCMV infection. Taking in account these observations, an aspect that deserves to be explored is whether these ICOSL-related activities can be employed by additional herpesviruses or other viral pathogens as a mean to escape host immune surveillance.

In conclusion, in this study, we present a novel viral tactic to impair T-cell activation, based on the interference with the ICOS:ICOSL co-stimulation pathway. To this end, the herpesviruses analyzed so far seem to exploit 'self' proteins, evolved during co-evolution with their hosts, being in the case of murine CMV, a multitasking immunoevasin. Ultimately, the study of these immunoevasins should provide not only new insights into viral pathogenesis, but also enlighten us about critical biological and immunological processes that promote viral control.

## Materials and methods

### Cell cultures

NIH3T3 (mouse embryonic fibroblasts), SVEC4-10 (SV40-transformed mouse endothelial cells), HFF (human foreskin fibroblasts), HEL299 (human embryonic lung fibroblasts), MEFs (primary mouse embryonic fibroblasts), MEFs immortalized with p53 (kindly provided by Dr. Jay Nelson [Oregon Health Sciences University, Portland, USA]), and Vero cells were cultivated in Dulbecco's modified Eagle's medium (DMEM) supplemented with 2 mM glutamine, 1 mM sodium pyruvate, 50 U/mL of penicillin, 50 g/mL of streptomycin, and 10% fetal bovine serum (FBS). IC-21 (mouse peritoneal macrophages transformed with SV-40), DC2.4 (mouse dendritic), NS1 (mouse myeloma), THP-1 (human monocytic), and 300.19 (mouse pre-B) cells were cultured in RPMI-1640 medium, supplemented as indicated above, and for THP-1 and 300.19 cells, 0.05 mM 2-mercaptoethanol was added. Primary mouse peritoneal macrophages were elicited from peritoneal exudate cells from BALB/c mice

receiving 1 mL of thioglycolate i.p. 4 days before the extraction. Primary peritoneal DCs were isolated according to *Ray and Dittel, 2010*. Splenocytes and lymph nodes obtained from mice were subjected to a manual disaggregation and treatment with red blood cell lysis buffer (0.15 M NH4Cl, 0.01 M Tris HCL) to obtain cell suspensions. Primary mouse DCs and macrophages from bone marrow of femur bones from BALB/c mice or C57BL/6 were obtained after 7 days of culture in RPMI-1640 medium supplemented as indicated above, with the addition of granulocyte-macrophage colony-stimulating factor (GM-CSF) at days 3 and 6 from extraction. Attached cells were enriched in macrophages, whereas suspension fractions were enriched in DCs. Both type of cells were plated in flat-bottom 24-well dishes with the supplemented RPMI-1640 medium at a concentration of $2 \times 10^5$ cells/well. Dendritic and macrophage differentiation were confirmed by flow cytometry using the markers CD11c and F4/80, respectively. Primary human monocytes were obtained from PBMCs isolated by Ficoll-Paque density-gradient centrifugation from fresh blood samples of human healthy donors. Monocyte-derived macrophages (MDM) were differentiated after culturing monocytes for 5 days in RPMI-1640 medium supplemented as indicated for mouse primary cells, with 50 ng/mL of GM-CSF. Cell lines were routinely tested for mycoplasma contamination with MycoAlert Mycoplasma Detection Kit from Lonza.

## Viruses and infections

The bacterial artificial chromosome (BAC)-derived MCMV strain MW97.01, here referred as MCMVwt (*Wagner et al., 1999*), and the MCMV-GFP recombinant virus, a derivative of MW97.01 carrying the GFP gene (*Mathys et al., 2003*), were used as parental viruses throughout the study. Recombinant strains MCMV-GFPΔ15 lacking genes from *m128* to *m138* (termed here MCMV-GFPΔm128-m138) and MCMVΔm138 lacking the *m138* gene, have been described previously (*Lenac et al., 2006*), as well as recombinant strain MCMV-GFPΔ1 lacking genes from *m1* to *m17* (termed here MCMV-GFPΔm1-m17; *Brune et al., 2006*). MCMV mutants containing m138 versions without Ig2 domain (MCMVm138ΔIg2), or without Ig2 and Ig3 domains (MCMVm138ΔIg2/Ig3) were also described in *Lenac et al., 2006*. Viral stocks were grown by infecting MEFs at a low moi using DMEM supplemented as stated before, except that 3% FBS was used. Cell supernatants were recovered when maximum cytopathic effect was reached, and cleared of cellular debris by centrifugation at 1750 g for 10 min. The HCMV strain TB40/E carrying the *GFP* gene was used in the study (termed here HCMV-GFP; *Sinzger et al., 2008*). HCMV stocks were prepared as indicated for MCMVs, but using in this case HEL299 or HFF cells. HSV-1 and HSV-2 containing the GFP gene were kindly provided by Dr. Antonio Alcamí (Autonomous University of Madrid, Spain). Both viruses were grown in Vero cells. Viral titers were determined by standard plaque assay on primary BALB/c MEFs, MEFs p53, HEL299 or Vero cells for MCMV, HCMV, or HSV-1 and HSV-2, respectively. Infections of murine peritoneal macrophages, BMDCs and IC-21 cells included a centrifugal enhancement of infectivity step (*Hudson, 1988*). C57BL/6 derived DCs were infected by incubating cells in suspension with the virus for 30 min in a small volume (c = $10^7$ cells/mL) with occasional agitation at 37°C. THP-1 cells were activated for 24 hr with 50 ng/mL PMA (Sigma-Aldrich) prior the infection. THP-1 cells and MDM were infected in the presence of PMA or GM-CSF, respectively, for 16 hr, and also included a step of centrifugal enhancement of infectivity. UV-inactivation of virus was performed using a UV crosslinker (HL 2000 hybrilinker) for 3 min at 360 mJ/cm$^2$.

## Antibodies

For flow cytometry analysis, we used the following anti-mouse antibodies: ICOSL-biotin or ICOSL-PE (clone HK5.3), MHCI-PE (clone M1/42), B7.1-APC (clone 16-10A1), CD84-PE (clone mCD84.7), CD11c-A647 or -APC (clone N418), F4/80-PE (clone BM8), B220-PB (clone RAE-6B2), CD5-PE (clone 53–7.3), CD138-PECy7 (clone 281–2), CD3-A488 (clone 17A2), CD4-PB (clone GK1.5), CXCR5-APC (clone L138D7), and CD19-A647 (clone 6D5), purchased from Biolegend; CD23-PE (clone B3B4), CD95-A647 (clone Jo2), CD44-FITC (clone IM7), and CD8-PE (clone 53–6.7), obtained from BD Biosciences; CD62L-APC (clone MEL-14), CD44-APC or -AlexaFluor700 (clone IM7), CD21-PECy7 (clone 8D9), GL7-A488 (clone GL7), CD3-PerCP-Cy5.5 or -PE-eFluor610 (clone 145–2 C11), ICOSL-PE (clone HK5.3), B7.1-APC (clone 16-10A1), CD3-PE-eFluor610 (clone 17A2), CD8-APC, -SuperBright780 or -FITC (clone 53–6.7), CD4-FITC (clone GK1.5), CD45.1-SuperBright600 (clone A20), CD45.2-eFluor506 (clone 104), CD19-PE-eFluor610 (clone eBio1D3), NKp46-PE-eFluor610 (clone 29A1.4),

CD11b-PE-Cy7 (clone M1/70), CD11c-PerCP-Cy5.5 (clone N418), MHC II-eFluor450 or –APC-eFluor780 (clone M5/114.15.2), TCR V alpha 11-APC (clone RR8-1) and IFN-γ (clone XMG1.2), from eBioscience, Thermo Fisher Scientific; PD-1-PE (clone J43.1) and CD3-PECy7 (clone 145–2 C11), from Tonbo; and IgM-FITC (polyclonal), from Southern Biotech. Armenian hamster IgG (clone eBio299 Arm) and rat IgG2a kappa (clone eBR2a) isotype controls were purchased from Thermo Fisher Scientific. The rat anti-mouse ICOSL (clone HK5.3), anti-mouse NK1.1 (clone PK136), anti-mouse CD8 (clone YTS 169.4), and anti-mouse CD4 (clone GK1.5) used for the in vivo assays were obtained from Bioxcell. The anti-mouse ICOSL (clone 599841) used for immunochemistry was purchased from Novusbio Biologicals. Biotin anti-human ICOSL (clone MIH12), CD4-APC (clone RPA-T4), and CD70 (clone 2F2) were purchased from Thermo Fisher Scientific, BD Biosciences, and Immunotools, respectively. Anti-human ICOSL (polyclonal), anti-actin (clone C4), anti-HA (clone C2974), anti-GFP (polyclonal), and anti-human IE1 (clone 8B1.2), used for Western blots, were from Elabscience, MP Biomedicals, Cell Signaling MP, Abcam, and Merck Millipore, respectively. The anti-mouse IE1 (named Croma101) and the anti-human Fc IgG (clone 29.5; Fc specific) were previously described (*Trgovcich et al., 2000*; *Pérez-Carmona et al., 2015*). Anti-mouse m04 (clones m04.17 or m04.16) antibody used as a marker of MCMV infection was produced in the Center for Proteomics (Faculty of Medicine, University of Rijeka, Croatia) and was described previously (*Železnjak et al., 2019*). The anti-mouse IgG-A555 was purchased from Life Technologies, the anti-rat IgG-A488 and anti-mouse IgG-PE or -A647 from Jackson ImmunoResearch, the anti-mouse IgG1-PerCP-eFluor710 and anti-mouse IgG2b-FITC from eBioscience, Thermo Fisher Scientific, the anti-rabbit IgG-HRP from Promega and the anti-mouse IgG-HRP from Sigma-Aldrich. The streptavidin-PE, -A555, or -HRP conjugates were purchased from BD Biosciences, Thermo Fisher Scientific, and Roche, respectively. The anti-m138 mAb (clone m138.1.120) was generated by fusing an NS1 myeloma cell line with spleen cells from a BALB/c mouse immunized three times with 300.19 cells expressing at the cell surface HA-m138. The m138.1.120 mAb was subcloned and purified using an Affi-Gel protein A MAPS II kit (Bio-Rad) as indicated before (*Engel et al., 2011*). Fixable viability dye eFluor780 and eFluor506 (1000x, eBioscience, Thermo Fisher Scientific) were used to stain dead cells.

## Plasmid constructions

HA-m138, containing the N-terminal HA-tagged m138 molecule without its signal peptide, was constructed by PCR using as a template DNA extracted from MCMV virions and primers m138PstlFor and m138PstlRev. The resulting PCR product was inserted into the pGEM-T vector, and subsequently, the *m138* fragment was excised with *Bgl*II and inserted in frame with the HA tag at the N-terminal end into the mammalian expression vector pDisplay (Invitrogen) opened with *Pst*I. The m138-GFP plasmid was obtained by PCR employing primers m138BamFor and m138BamRev, containing a *Bam*HI restriction site, and DNA from MCMV virions as a template. The PCR product was inserted into the pGEM-T vector, and then the m138 fragment without its terminal codon was excised with *Bam*HI and inserted in frame with the GFP at the C-terminal end into the pEGFP-N3 plasmid (clontech) opened with *Bam*HI. m138ΔIg1-GFP, without the first Ig domain of m138 (322nt-681nt), m138ΔIg2-GFP, without the second Ig domain of m138 (697nt-1029nt), and m138ΔIg2/3-GFP, without the second and third Ig domains of m138 (697nt-1416nt), were constructed as follows: first, independent PCR products were generated using as a template the pEGFP-N3 m138-GFP vector, two common external primers (m138BamFor and m138BamRev), and for each construct, two internal complementary primers bearing the corresponding deletions: SOEm138ΔIg1Rev and SOEm138ΔIg1For for m138ΔIg1; SOEm138ΔIg2Rev and SOEm138ΔIg2For for m138ΔIg2; and SOEm138ΔIg2/3Rev and SOEm138ΔIg2/3For for m138ΔIg2/3. Splicing by overhang extension (SOE)-PCR was then performed to join the two PCR fragments obtained for each m138 mutant using the external primers. These new PCR products were inserted into the pGEM-T vector, and then fragments of the m138 mutants without their terminal codon stop were excised and cloned in frame with the GFP at the C-terminal end into the pEGFP-N3 plasmid opened with *Bam*HI. Mouse ICOSL, without its signal peptide, fused to the HA epitope at the N-terminus (HA-ICOSL) was obtained by PCR employing as a template pCMV6-mouse ICOSL vector (Origene) and primers ICOSLmurineSall-For and ICOSLmurineNotlRev. The resulting PCR product was inserted into the pGEM-T vector and a *Sal*I-*Not*I fragment corresponding to the murine ICOSL was then excised and inserted into the mammalian expression vector pDisplay opened with the same restriction sites. In all cases, the PCR products inserted in the pGEM-T vector were sequenced to validate the nucleotide sequences, and

when required, the correct orientations of the cloned inserts were assessed employing different restriction sites.

## Flow cytometry analysis

Flow cytometry was performed according to the Guidelines for the use of flow cytometry and cell sorting in immunological studies (*Cossarizza et al., 2019*). To minimize non-specific staining, all incubations were carried out in the presence of 20% rabbit serum (Linus) and 1% of fetal bovine serum in PBS. Adherent cells were harvested by incubation with 2 mM EDTA or when indicated by treatment with trypsin at 37°C. For intracellular staining, cells were fixed and permeabilized with either the Intracellular fixation and permeabilization buffer set or Foxp3 staining buffer set (eBioscience, Thermo Fisher Scientific) following manufacturer's instructions, before antibody incubations. Prior to cytometry analysis cell suspensions from spleens, lungs, salivary glands, and salivary gland draining lymph nodes were washed and filtered through a 70-µm-cell strainer (Biologix). All samples were acquired with LSRII Fortessa, FACSAria III or FACSCanto II flow cytometers (BD Biosciences) and analyzed with FlowJo Xv10.0.7 (Tree Star, Inc) software. At least two independent experiments were carried out for each subject of analysis.

## Transient and stable transfections

NIH3T3 cells were transiently transfected with 2 µg of the corresponding plasmid using the Amaxa Cell Line Nucleofactor Kit R according to the manufacturer's protocol. Likewise, 300.19 and IC-21 cells were transfected employing the Amaxa Cell Line Nucleofector Kit V. To generate stable 300.19 cell lines expressing HA-m138 or NIH3T3 expressing HA-ICOSL, transfections were followed by G418 (Invitrogen) selection.

## Western blot analysis

NIH3T3 and THP-1 cells were harvested by scrapping and extracts prepared in lysis buffer (20 mM Tris-HCl pH 7.5, 1 mM EDTA, 150 mM NaCl, and 1% Triton X-100) supplemented with protease inhibitors (1 mM Na3VO4, 1 mM PMSF, 0.6 µg/mL Aprotinine, 2 µg/mL Leupeptin, and 312 µg/mL Benzamidin). When specified, cells were previously incubated with 250 µM leupeptin and 20 nM bafilomycin A1 (Sigma). Lysates were quantified by a bicinchoninic acid (BCA) protein assay kit (Thermo-Scientific), and 15 µg of proteins were subjected to SDS-PAGE in 10% acrylamide gels and subsequently transferred to nitrocellulose membranes (Protran). The membranes were incubated with the indicated antibodies and blots were developed using the SuperSignal West Pico chemiluminescent substrate (Pierce) according to the manufacturer's protocol. Anti-actin mAb was used as a loading control, and anti-IE1 mAbs were used as infection controls. At least two independent experiments were carried out for each subject of analysis.

## Immunoprecipitation

NIH3T3 cells non-trasfected or transfected with HA-ICOSL were surface labeled with biotin (Sigma-Aldrich) and lysed. Cell lysates were precleared three times for 30 min employing protein G-Sepharose (GE Healthcare) and immunoprecipitated by incubation with an anti-HA agarose conjugate (Sigma-Aldrich). Samples from immunoprecipitates were subjected to SDS-PAGE and western blot analysis. Membranes were incubated with streptavidin-HRP or with anti-HA followed by anti-rabbit IgG-HRP. At least two independent experiments were carried out for each subject of analysis.

## Co-immunoprecipitation

NIH3T3 cells co-transfected with m138-GFP and HA-ICOSL were lysed with a soft lysis buffer (20 mM Tris HCl pH 8, 100 mM NaCl, 2 mM EDTA, and 0.5% Triton-X-100) supplemented with Halt protease inhibitor cocktail, during 30 min at 4°C. The lysates were incubated with G-Sepharose beads in three rounds for 30 min each, followed by an overnight incubation with an anti-HA agarose conjugate. The immunoprecipitates were washed four times with Co-IP lysis buffer before the samples were subjected to western blot analysis. Membranes were incubated with anti-GFP or with anti-HA, followed by an anti-rabbit IgG-HRP. At least two independent experiments were carried out for each subject of analysis.

## Immunofluorescence microscopy

NIH3T3 cells and 300.19 cells, either untransfected or stably transfected with HA-ICOSL or HA-m138, respectively, were cultured on glass coverslips in 24-well tissue culture plates. When specified, NIH3T3 cells were MCMV infected and exposed to lysosomal inhibitors as indicated for the Western blot analysis. At 24 hpi, cells were washed in PBS and fixed and permeabilized using 4% paraformaldehyde and 0.05% Triton X-100 (for intracellular staining), or just fixed in paraformaldehyde (for cell surface detection), and subsequently blocked with PBS 6% FBS. Cells were stained with anti-m138 mAb or anti-ICOSL, using as secondary antibodies an anti-mouse IgG-A555 or an anti-rat IgG-A488. Nuclei were counterstained with DAPI reagent (Invitrogen). In assays where the LysoTracker Red DND-99 (Molecular Probes) was used, cells were incubated at 37°C for 2 hr with 100 nM of the fluorescence dye, fixed with 4% paraformaldehyde, gently permeabilized with 0.02% saponin, and immediately processed as indicated above. In this case, the anti-m138 mAb and an anti-mouse IgG-A488 were used. The samples were mounted in ProLong Gold antifade reagent (Invitrogen). Cells were examined under a fluorescence microscope at 405 nm (DAPI), at 555–565 nm (A555), and at 495–518 (A488). GFP was observed at 450–490. Fluorescence images were obtained using a Nikon Eclipse E600 microscope (Nikon) or an inverted Leica DMI6000B microscope and the LAS AF software from Leica Microsystems (Leica, Wetzlar). A Zeiss LSM880 laser scanning spectral confocal microscope (Zeiss, Jena) equipped with an Axio Observer seven inverted microscope, blue diode (405 nm), Argon (488 nm), diode pumped solid state (561 nm) and HeNe (633 nm) lasers and a Plan Apochromat 63x oil (NA 1.4) immersion objective lenses was also used. DAPI, A-488 and A-555 images were acquired sequentially using 405, 488, and 561 laser lines, AOBS (Acoustic Optical Beam Splitter) as beam splitter and emission detection ranges 415–480, 500–550 nm, and 571–625 nm, respectively, and the confocal pinhole set at 1 Airy units. Spectral detection was performed using two photomultipliers and one central GaAsP detector. Analysis of the co-localization of m138 and the LysoTracker red fluorescence was determined using the ImageJ software analyzing 20 cells from each sample. Colocalization was assessed using Coloc2 plugin from FIJI/ImageJ program (*Schindelin et al., 2012*). A macro of instructions was created to process and automate colocalization quantification. Briefly, red and green channel images were background substracted with Rolling Ball Radius of 50 and colocalization was quantified from a region of interest delimiting the cell contour. Manders coefficients were analyzed from each image. M2 Manders colocalization coefficient indicates the percentage of m138 colocalizing with lysosomes, and M1 Manders coefficient the percentage of lysosomes colocalizing with m138. At least two independent experiments were carried out for each subject of analysis.

## Immunohistochemistry

Mice were euthanized at the indicated time points and spleens immediately collected within O.C.T. (Sakura) cryo-embedding media at −80°C. Consecutive tissue sections of 2.5 µm were obtained in the cryostat and fixed/permeabilized in acetone. Unspecific binding was prevented with PBS 6% FBS blocking solution. Antigens were detected with a rat anti-mouse CD45R/B220 and PNA (peanut lectin) biotin (Sigma-Aldrich). After 1 hr of incubation at room temperature, B220 antigen was revealed with anti-rat IgG- A488 and PNA with streptavidin-A555. Labeled tissue sections were visualized with a fluorescence microscope. Four consecutive tissue sections were evaluated from three representative mice of each experimental group. Follicle and GC areas were calculated from the total spleen sections of one representative mice of each group using ImageJ software. At least two independent experiments were carried out for each subject of analysis.

## Mouse infections, antibody treatment, and determination of viral titers

Four-week-old BALB/c mice (females and males) were obtained from Harlan (Netherlands) and housed in the vivarium of the Medical School of the University of Barcelona under specific-pathogen-free (SPF) conditions. Eight-to-twelve-week-old C57BL/6, TCR transgenic mice specific for M38 (Maxi mice; *Torti et al., 2011*), TCR transgenic mice specific for M25-II (*Mandaric et al., 2012*), C57BL/6 SCID and BALB/c mice were housed and bred under SPF conditions at the Central Animal Facility of the Medical Faculty of the University of Rijeka. When specified, ICOSL was blocked by i.p. injection of a rat anti-mouse ICOSL monoclonal antibody (clone HK5.3) at a concentration of 100 µg per mouse. The monoclonal antibody was administrated 1 day before infection and on days 2, 5, 8,

and 11 after infection. The efficacy of the in vivo ICOSL blockade was assessed by cytofluorometric analyses of spleen cells from treated mice using an anti-ICOSL recognizing the same epitope than the antibody used in the in vivo assay. For these assays, mice were i.p. inoculated with $2.5 \times 10^6$ PFU/mouse of tissue culture-propagated MCMVΔm138. Mice were observed and weighed during all the experiment and at 14 days post-infection, animals were sacrificed, and specific organs were removed, and their sera collected. Salivary gland and lungs were harvested as a 10% (weight/volume) tissue homogenate. Tissue homogenates were sonicated and centrifuged, and viral titers from the supernatants of individual mice were determined by standard plaque assays in MEFs, including a centrifugal enhancement of infectivity step. In experiments evaluating the in vivo effect of MCMV infection on ICOSL surface levels, mice were i.p. inoculated with $2 \times 10^6$ PFU/mouse of MCMV-GFP, and 2 days after infection, cells present in the peritoneal cavity were harvested with 5 mL of PBS and analyzed by flow cytometry. In depleting assays, 8-to-12-week-old C57BL/6 mice were infected i.v. with $2 \times 10^5$ PFU/mouse of MCMVwt, MCMVΔm138, or MCMVm138ΔIg2. On the day of infection and on days 5th and 10th after infection, indicated groups of mice were injected i.p. with 250 µg of anti-mouse NK1.1, 150 µg of anti-CD8, or 150 µg anti-mouse CD4 depleting antibodies. At day 14 post-infection, mice were sacrificed and viral titers in salivary glands and lungs of individual mice were determined by standard plaque assays. To obtain high-dose systemic infection, BALB/c mice were inoculated i.p. with $10^6$ PFU/mouse of MCMVwt or MCMVΔm138 for 6 hr after which peritoneal exudate cells were isolated according to *Ray and Dittel, 2010*. At least two independent experiments were carried out for each subject of analysis.

### Adoptive transfer assay

C57BL/6 SCID mice were i.p. infected with $2 \times 10^5$ PFU/mouse of MCMVwt or MCMVΔm138. One day prior to infection, mice were first injected i.p. with 250 µg of anti-mouse NK1.1 and a few hours later were given $10^5$ M25-II CD4$^+$ T cells. Another NK cell depletion followed on day 3 post-infection. Spleen, lungs and salivary glands were isolated 7 days after infection and the expansion of CD4$^+$ T cell (CD45.1$^+$ CD4$^+$) was analyzed. M25-II CD4$^+$ T cells were obtained from naive C57BL/6 transgenic M25-II mice. Briefly, splenocytes from C57BL/6 M25-II mice were isolated and enriched using CD4$^+$ T Cell Isolation Kit (Miltenyi). Percentage of M25-II CD4$^+$ T cells was obtained by staining with TCR V alpha 11 (clone RR8-1, eBioscience ThermoFisher) antibody.

### In vitro stimulation of CD8$^+$ T cells

C57BL/6 BMDCs were infected with 3 PFU/cell of indicated viruses. After 24 hr of infection, splenocytes from naive Maxi mice were added at the different T:E (BMDCs:CD8$^+$) ratios to infected BMDCs together with Brefeldin A (eBioscience, Thermo Fisher Scientific). After 6 hr of co-cultivation, IFN-γ production by CD8$^+$ T cells was measured. At least three independent experiments were carried out for each subject of analysis with two technical replicates/experiment.

### Quantification of serum anti-MCMV antibody levels

Detection of antibodies against MCMV on sera of MCMV-infected mice was performed by sandwich ELISA using 1.5 µg/well of MCMVΔm138-infected MEF lysate to coat the ELISA plates as described in *Miletic et al., 2017*. Briefly, $5 \times 10^6$ MEFs were infected with 0.01 PFU/cell with MCMVΔm138 until a high infection was reached after 5 days. Cells were then harvested by incubation with 2 mM EDTA and washed twice with cold PBS. Cells were sonicated in bicarbonate buffer, quantified by BCA and stored at −20°C. Diluted sera were incubated for 2 hr on the coated plates, followed by incubation with anti-mouse IgG-HRP to measure total IgGs, or with anti-mouse IgG1, IgG2a, IgG2b, IgG3, or IgM biotin conjugated (Jackson ImmunoResearch), followed by streptavidin-HRP. After washing, freshly prepared TMB substrate solution was added and the absorbance was measured at 450 and 570 nm wavelength (Thermo Scientific Multiskan FC). At least two independent experiments were carried out for each subject of analysis with at least two technical replicates/experiment.

### In vitro neutralization

Collected sera from MCMV infected mice were decomplemented at 56°C for 30 min, and then incubated, at the indicated dilution, with 100 PFU of the MCMV-GFP in DMEM containing or not 25% of rabbit serum for 1 hr at 37°C. The mixtures were then transferred to monolayers of MEFs p53 grown

in wells of 24-well plates, and further incubated 1 hr at 37°C. Next, cells were covered with medium containing 0.25% agarose and lysis plaques produced after 5–7 days in each condition counted. Results were represented as the percentage of plaque reduction, determined by the ratio of the amount of the plaques counted in the sample wells relative to the amount of plaques visualized in wells containing the serum of an uninfected mouse. At least two independent experiments were carried out for each subject of analysis with at least three technical replicates/experiment.

## Statistical analyses

Results are expressed as mean +/- standard deviation (SD) or standard error of the mean (SEM) of at least three independent experiments. For in vivo experiments, mice were pooled and randomized in experimental groups from 4 to 6 animals. The sample size was determined by pilot studies and according to the accepted practices in previous literature using the MCMV model. The selection of the appropriate statistical test was based on the number and distribution of data points per set. Differences between group means or individual values were assessed using the unpaired two-tailed *t*-test. Statistical differences between more than two study groups were evaluated using either the one-way ANOVA test or the Kruskal-Wallis bidirectional test. Exact *p*-values considered statistically significant and statistical details of experiments are indicated in the figure legends. All statistical analyses were performed using GraphPad Prism software version 7.03 and 8.1.1.

The key Resources Table (Appendix 1—key resource table 1) lists the key reagents used in this study.

## Acknowledgements

We thank Adriana Lázaro for technical assistance with the production of the monoclonal antibody against m138, Manuel Eduardo Sáez Moya for help in the preparation of histological tissue sections, and Francesc Poblador and Pablo Hernández-Luis for technical assistance in the immunohistochemistry assays. We acknowledge the use of the Advanced Optical Microscopy Facility at the University of Barcelona, and María Calvo for the analysis of confocal fluorescence microscopy images. We also thank Antonio Alcamí (Autonomous University of Madrid, Spain) for providing HSV-1 and HSV-2 expressing GFP. In addition, we thank Lea Hiršl, Tina Jenuš and Maja Cokarić Brdovčak for the help involving adoptive transfer assay and genotyping of C57BL/6 M25-II mice.

## Additional information

### Competing interests

Stipan Jonjic: Reviewing editor, *eLife*. The other authors declare that no competing interests exist.

### Funding

| Funder | Grant reference number | Author |
| --- | --- | --- |
| Ministerio de Economía y Competitividad | SAF 2017-87688 | Ana Angulo |
| Ministerio de Economía y Competitividad | RTI2018-094440-B-I00 | Pablo Engel |
| European Regional Development Fund | KK.01.1.1.01.0006 | Stipan Jonjic |
| University of Rijeka | Uniri-biomed-18-170 | Astrid Krmpotić |
| Croatian Science Foundation | Croatian-Swiss Research Program | Annette Oxenius Astrid Krmpotić |
| Swiss National Science Foundation | Swiss-Croatian Cooperation Programme | Annette Oxenius Astrid Krmpotić |
| Deutsche Forschungsgemeinschaft | Projektnummer 421451057 – FOR2830 | Martin Messerle Stipan Jonjic |
| Deutsche Forschungsge- | HE2526/9-1 | Hartmut Hengel |

meinschaft

| Ministerio de Economía y Competitividad | Formación de Personal Investigador fellowship | Guillem Angulo Joan Puñet-Ortiz |

The funders had no role in study design, data collection and interpretation, or the decision to submit the work for publication.

## Author contributions

Guillem Angulo, Data curation, Formal analysis, Investigation; Jelena Zeleznjak, Formal analysis, Methodology; Pablo Martínez-Vicente, Data curation, Formal analysis, Investigation, Methodology; Joan Puñet-Ortiz, Formal analysis, Investigation, Methodology; Hartmut Hengel, Martin Messerle, Annette Oxenius, Resources; Stipan Jonjic, Pablo Engel, Conceptualization, Resources, Supervision, Funding acquisition; Astrid Krmpotić, Conceptualization, Supervision, Funding acquisition, Methodology; Ana Angulo, Conceptualization, Resources, Supervision, Funding acquisition, Visualization, Writing - original draft, Project administration, Writing - review and editing

## Author ORCIDs

Guillem Angulo https://orcid.org/0000-0001-7086-9754
Jelena Zeleznjak http://orcid.org/0000-0001-6619-3675
Pablo Martínez-Vicente https://orcid.org/0000-0001-9277-1950
Hartmut Hengel http://orcid.org/0000-0002-3482-816X
Pablo Engel https://orcid.org/0000-0001-8410-252X
Ana Angulo https://orcid.org/0000-0002-5792-1164

## Ethics

Human subjects: Human blood was obtained from healthy volunteer donors through the Blood and Tissue Bank of the Catalan Department of Health (Barcelona, Spain). Utilization of blood products for the experiments conducted was approved by the Ethics Committee of the Hospital Clinic of Barcelona (Barcelona, Spain), and according to the principles of the Declaration of Helsinki.

Animal experimentation: All procedures involving animals and their care were approved (protocol number CEEA 308/12) by the Ethics Committee of the University of Barcelona (Spain) and the Animal Welfare Committee at the University of Rijeka (Croatia) and were conducted in compliance with institutional guidelines as well as with national (Generalitat de Catalunya decree 214/1997, DOGC 2450) and international (Guide for the Care and Use of Laboratory Animals, National Institutes of Health, 85-23, 1985) laws and policies.

## Decision letter and Author response

Decision letter https://doi.org/10.7554/eLife.59350.sa1
Author response https://doi.org/10.7554/eLife.59350.sa2

# Additional files

## Supplementary files

• Transparent reporting form

## Data availability

All data generated or analyzed during this study are included in the manuscript and supporting files.

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

# Appendix 1

**Appendix 1—key resources table**

| Reagent type (species) or resource | Designation | Source or reference | Identifiers | Additional information |
|---|---|---|---|---|
| Antibody | anti-mouse CD8 (clone YTS 169.4) (Rat monoclonal) | Bio X cell West Lebanon, NH, USA | Cat#: BE0117 RRID:AB_10950145 | in vivo depletion (150 µg/injection) |
| Antibody | anti-mouse CD4 (clone GK1.5) (Rat monoclonal) | Bio X cell West Lebanon, NH, USA | Cat#: BE0003-3 RRID:AB_1107642 | in vivo depletion (150 µg/injection) |
| Antibody | anti-mouse NK.1.1 (clone PK136) (Rat monoclonal) | Bio X cell West Lebanon, NH, USA | Cat#: BE0036 RRID:AB_1107737 | in vivo depletion (250 µg/injection) |
| Antibody | anti-mouse ICOSL (clone HK5.3) (Rat monoclonal) | Bio X cell West Lebanon, NH, USA | Cat#: BE0028 RRID:AB_1107566 | in vivo (100 µg/injection) |
| Antibody | anti-m04 (clone m04.17) (Mouse monoclonal) | Center for Proteomics, Faculty of Medicine, University of Rijeka, Croatia PMID:31142589 | | Stock 1 mg/mL FACS (1:100) |
| Antibody | anti-m04 (clone m04.16) (Mouse monoclonal) | Center for Proteomics, Faculty of Medicine, University of Rijeka, Croatia PMID:31142589 | | Stock 1 mg/mL FACS (1:100) |
| Antibody | anti-HA (clone C2974) (Mouse monoclonal) | Sigma-Aldrich, St. Louis, MO, USA | Cat#: H9658 RRID:AB_260092 | WB (1:2000) |
| Antibody | anti-human IE1 (clone 8B1.2) (Mouse monoclonal) | Merck Millipore, Burlington, MA, USA | Cat#: MAB810R RRID:AB_11212266 | WB (1:2000) |
| Antibody | anti-human Fc IgG (clone 29.5) (Mouse monoclonal) | In house Department of Biomedical Sciences, University of Barcelona, Spain | | Stock 1 mg/mL FACS (1:400) |
| Antibody | anti-m138 (clone m138.1.120) (Mouse monoclonal) | In house Department of Biomedical Sciences, University of Barcelona, Spain | This manuscript | Stock 1 mg/mL FACS (1:100) IF (1:50) |
| Antibody | anti-mouse CD80 (clone 16-10A1) (Armenian hamster monoclonal) | Biolegend San Diego, CA, USA | Cat#: 104714 RRID:AB_313134 | FACS (1:200) |
| Antibody | anti-human ICOSL (clone MIH12) (Mouse monoclonal) | Thermo Fischer Scientific, Waltham, MA, USA | Cat#: 16-5889-82 RRID:AB_469129 | FACS (1:50) |

*Continued on next page*

*Appendix 1—key resources table continued*

| Reagent type (species) or resource | Designation | Source or reference | Identifiers | Additional information |
|---|---|---|---|---|
| Antibody | anti-mouse ICOSL (clone HK5.3) (Rat monoclonal) | Biolegend San Diego, CA, USA | Cat#: 107403 RRID:AB_345259 (Biotin) Cat#: 107405 RRID:AB_2248797 (PE) | FACS (1:100) IF (1:50) |
| Antibody | anti-human ICOSL (Rabbit polyclonal) | Elabscience Wuhan, Hubei, China | Cat#: E-AB-15519 | WB (1:1000) |
| Antibody | anti-mouse ICOSL (clone 599841) (Rat monoclonal) | R and D Systems, Minneapolis, MN, USA | Cat#: MAB158 RRID:AB_10719415 | IF (1:500) |
| Antibody | anti-m123/IE1 (MCMV) (clone IE1.01) (Mouse monoclonal) | Center for Proteomics, Faculty of Medicine, University of Rijeka, Croatia | Cat#: HR-MCMV-12 | Stock 1 mg/mL IF (1:100) FC (1:100) WB (1:1000) |
| Antibody | anti-HA—Agarose conjugate (Mouse monoclonal) | Sigma-Aldrich, St. Louis, MO, USA | Cat#: A2095 RRID:AB_257974 | |
| Cell line (*M. musculus*) | NIH3T3 | ATCC, Manassas, VA, USA | Cat#: CRL-1658 RRID: CVCL_0594 | |
| Cell line (*M. musculus*) | DC2.4 | Merck Millipore, Burlington, MA, USA | Cat#: SCC142 RRID: CVCL_J409 | |
| Cell line (*M. musculus*) | MEFs p53 | Dr. Jay Nelson, Health Sciences University, Oregon, USA | | |
| Cell line (*M. musculus*) | SVEC4-10 | ATCC, Manassas, VA, USA | Cat#: CRL-2181 RRID: CVCL_4393 | |
| Cell line (*M. musculus*) | IC-21 | ATCC, Manassas, VA, USA | Cat#: TIB-186 RRID: CVCL_3726 | |
| Cell line (*M. musculus*) | NS1 | ECACC, Public Health England, Salisbury, UK | Cat#: 85011427 RRID: CVCL_2155 | |
| Cell line (*M. musculus*) | 300.19 | Dr. Michel Streuli, Dana Farber Cancer Institute, MA, USA | | |

*Continued on next page*

*Appendix 1—key resources table continued*

| Reagent type (species) or resource | Designation | Source or reference | Identifiers | Additional information |
|---|---|---|---|---|
| Cell line (*Homo-sapiens*) | THP-1 | ATCC, Manassas, VA, USA | Cat#: TIB-202 RRID: CVCL_0006 | |
| Cell line (*Homo-sapiens*) | HFF | ATCC, Manassas, VA, USA | Cat#: SCRC-1041 RRID: CVCL_3285 | |
| Cell line (*Homo-sapiens*) | HEL299 | ATCC, Manassas, VA, USA | Cat#: CCL-137 RRID: CVCL_2480 | |
| Cell line (*Cercopithecus aethiops*) | Vero | ECACC, Public Health England, Salisbury, UK | Cat#: 84113001 RRID: CVCL_0059 | |
| Cell line (*Cercopithecus aethiops*) | COS-7 | ATCC, Manassas, VA, USA | Cat#: CRL-1651 RRID: CVCL_0224 | |
| Strain, strain background-(*Mouse cytomegalovirus*) | wild type (BAC-derived strain pSM3fr) | PMID:10400809 | | |
| Strain, strain background-(*Mouse cytomegalovirus*) | BAC-derived strain pSM3fr-GFP | PMID:12660946 | | |
| Strain, strain background-(*Mouse cytomegalovirus*) | Mutant MCMV-GFPΔ15 | PMID:16831899 | | Lacks ORFs m128-m138 |
| Strain, strain background-(*Mouse cytomegalovirus*) | Mutant MCMV-GFPΔm138 | PMID:16831899 | | Lacks ORF m138 |
| Strain, strain background-(*Mouse cytomegalovirus*) | Mutant MCMV-GFPΔ1 | *Brune et al., 2006* | | Lacks ORFs m1-m17 |
| Strain, strain background-(*Mouse cytomegalovirus*) | Mutant MCMV-m138ΔIg2 | PMID:16831899 | | Lacks Ig2 domain of m138 |
| Strain, strain background-(*Mouse cytomegalovirus*) | Mutant MCMV-m138ΔIg2/Ig3 | PMID:16831899 | | Lacks Ig2 and Ig3 domains of m138 |
| Strain, strain background-(*Human cytomegalovirus*) | HCMV-GFP (strain TB40/E) | PMID:18198366 | | |

*Continued on next page*

*Appendix 1—key resources table continued*

| Reagent type (species) or resource | Designation | Source or reference | Identifiers | Additional information |
|---|---|---|---|---|
| Strain, strain background (*M. musculus*) | MAXI mice | Central Animal Facility, Faculty of Medicine, University of Rijeka, Croatia PMID:22046127 | | |
| Strain, strain background (*M. musculus*) | BALB/c mice | Central Animal Facility, Faculty of Medicine, University of Barcelona, Spain and Central Animal Facility, Faculty of Medicine, University of Rijeka, Croatia | | |
| Strain, strain background (*M. musculus*) | C57BL/6 mice | Central Animal Facility, Faculty of Medicine, University of Rijeka, Croatia | | |
| Strain, strain background (*M. musculus*) | M25-II mice | Central Animal Facility, Faculty of Medicine, University of Rijeka, Croatia PMID:22876184 | | |
| Strain, strain background (*M. musculus*) | C57BL/6 SCID | Central Animal Facility, Faculty of Medicine, University of Rijeka, Croatia | | |
| Chemical compound, drug | Amaxa Cell Line Nucleofactor Kits R and V | Lonza, Basel, Switzerland | Cat#: VCA-1001 and VCA-1003 | |
| Chemical compound, drug | G418 | Invivogen, Toulouse, France | Cat#: ant-gn-1 | 1.2 mg/mL |
| Chemical compound, drug | leupeptin | Sigma-Aldrich, St. Louis, MO, USA | Cat#: L2884 | 250 µM |
| Chemical compound, drug | bafilomycin A1 | Sigma-Aldrich, St. Louis, MO, USA | Cat#: B1793 | 20 nM |
| Chemical compound, drug | Biotin | Sigma-Aldrich, St. Louis, MO, USA | Cat#: B5161 | 2 mg/ million of cells |
| Chemical compound, drug | Protein G Sepharose 4 Fast Flow | GE Healthcare, Chicago, IL, USA | Cat#: 17-0618-01 | |
| Chemical compound, drug | Tissue-Tek O.C.T. Compound | Sakura, Torrance, CA, USA | Cat#: 4583 | |
| Chemical compound, drug | PNA biotin | Sigma-Aldrich, St. Louis, MO, USA | Cat#: L6135 | |
| Chemical compound, drug | Brefeldin A | eBioscience, Thermo Fischer Scientific, Waltham, MA, USA | Cat#: 00-4506-51 | 1 µg/ml |
| Chemical compound, drug | LysoTracker Red DND-99 | Invitrogen, Thermo Fischer Scientific, Waltham, MA, USA | Cat#: L7528 | |
| Chemical compound, drug | Intracellular Fixation and Permeabilization Buffer Set | eBioscience, Thermo Fischer Scientific, Waltham, MA, USA | Cat#: 88-8824-00 | |

*Continued on next page*

*Appendix 1—key resources table continued*

| Reagent type (species) or resource | Designation | Source or reference | Identifiers | Additional information |
|---|---|---|---|---|
| Chemical compound, drug | Foxp3/ Transcription Factor Fixation/Permeabilization | eBioscience, Thermo Fischer Scientific, Waltham, MA, USA | Cat#: 00-5521-00 | |
| Cell Isolation kit | CD4$^+$ T Cell Isolation Kit, mouse | Miltenyi Biotec, Bergisch Gladbach, Germany | Cat#: 130-104-454 | |
| Recombinant DNA reagent | pGEM-T (plasmid) | Promega, Madison, WI, USA | Cat#: A3600 | |
| Recombinant DNA reagent | pDisplay (plasmid) | Invitrogen, Thermo Fischer Scientific, Waltham, MA, USA | Cat#: V660-20 | |
| Recombinant DNA reagent | pEGFP-N3 (plasmid) | Clontech, Takara, Tokio, Japan | Cat#: 6080–1 | |
| Recombinant DNA reagent | pCMV6-mouse ICOSL (plasmid) | Origene, Rockville, MD, USA | Cat#: MR204667 | |
| Sequence-based reagent | m138PstIFor: 5'-CTGCAGGCATCAATTACCTGCGTGCCAG-3' | Sigma-Aldrich, St. Louis, MO, USA | | |
| Sequence-based reagent | m138PstIRev: 5'-CTGCAGTTACGTGTGACGTACGCAACC-3' | Sigma-Aldrich, St. Louis, MO, USA | | |
| Sequence-based reagent | m138BamFor: 5'-GGATCCATGGCGCCTTCGACGCTGATC-3' | Sigma-Aldrich, St. Louis, MO, USA | | |
| Sequence-based reagent | m138BamRev: 5'-GGATCCCGTGTGACGTACGCAACCCGG-3' | Sigma-Aldrich, St. Louis, MO, USA | | |
| Sequence-based reagent | SOEm138ΔIg1Rev: 5'-AGTCCCGGTGGAGTCGGTGATCAGTTGCGT-3' | Sigma-Aldrich, St. Louis, MO, USA | | |
| Sequence-based reagent | SOEm138ΔIg1For: 5'-ACGCAACTG ATCACCGACTCCACCGGGACT-3' | Sigma-Aldrich, St. Louis, MO, USA | | |
| Sequence-based reagent | SOEm138ΔIg2Rev: 5'-CATCAGCG AC CGGCCAGT CCCGGTGGAGTC-3' | Sigma-Aldrich, St. Louis, MO, USA | | |
| Sequence-based reagent | SOEm138ΔIg2For: 5'-GACTCCACC GGGACTGGCCGGTCGCTGATG-3' | Sigma-Aldrich, St. Louis, MO, USA | | |
| Sequence-based reagent | SOEm138ΔIg2/3Rev: 5'-CTGAG GGGACGTGACAGTCCCGGTGGAGTC-3' | Sigma-Aldrich, St. Louis, MO, USA | | |
| Sequence-based reagent | SOEm138ΔIg2/3For: 5'-GACTC CACCGGGACTGTCACGTCCCCTCAG-3' | Sigma-Aldrich, St. Louis, MO, USA | | |
| Sequence-based reagent | ICOSLmurineSalIFor: 5'-GTCGACAGAGACTGAAGTCGGTGCAATG-3' | Sigma-Aldrich, St. Louis, MO, USA | | |
| Sequence-based reagent | ICOSLmurineNotIRev: 5'-GCGGCC GCTTAGGCGTGGTCTGTAAGTTCA-3' | Sigma-Aldrich, St. Louis, MO, USA | | |

