## [Decision Letter]

**Acceptance summary:**

Viruses are experts as figuring ways to evade the immune system. In this study, the authors identify a previously undescribed viral immune evasion strategy in which herpesviruses interfere with the proper expression of a co-stimulatory molecule called ICOSL. This results in impaired T cell responses and reduced ability to control viral replication in vivo.

**Decision letter after peer review:**

Thank you for submitting your article "Cytomegalovirus restricts ICOSL expression on APCs disabling T cell costimulation and contributing to immune evasion" for consideration by *eLife*. Your article has been reviewed by three peer reviewers, one of whom is a member of our Board of Reviewing Editors, and the evaluation has been overseen by Satyajit Rath as the Senior Editor. The following individual involved in review of your submission has agreed to reveal their identity: Ian R Humphreys (Reviewer #2).

The reviewers have discussed the reviews with one another and the Reviewing Editor has drafted this decision to help you prepare a revised submission.

Summary:

In the manuscript, "Cytomegalovirus restricts ICOSL expression on APCs disabling T cell co-stimulation and contributing to immune evasion," Angulo et al. identify a new viral immune evasion strategy in which herpesviruses interfere with T cell stimulation by antagonizing with the ICOS-ICOSL co-stimulatory activation pathway. The primary focus of the study in murine cytomegalovirus where the authors identify that m138 targets ICOSL for lysosomal degradation and this facilitates virus persistence in vivo. They go on to demonstrate that HCMV and HSV also target ICOSL and, in the case of HCMV, provide evidence supporting a role for the HCMV US16 protein in targeting ICOSL.

Essential revisions:

1) Strengthen lysosomal degradation and ICOSL- m138 localisation:

a) Reviewer #1: While the authors cite previous studies that colocalize LAMP1 and m138, those studies used different cell systems and did not actually quantify colocalization with Pearson coefficients. A key mechanistic argument of this study is m138-mediated targeting of ICOSL to lysosomes. It would seem appropriate for the authors to establish and quantify this colocalization in their specific system. (Additionally, the Thale 1994 paper referenced does not appear to match the statement its associated with.)

b) Reviewer #2: Given the focus of DCs in this study, why did the authors use NIH 3T3 cells for the localizations studies? Some justification should be given and/or key observations should be repeated in these cell types.

2) Clarify leupeptin/Bafilomycin blocking studies:

a) Reviewer #1: In Figure 4G,4H, the authors claim that Leu+Baf results in "significant increase of the 51kDa band". An increase is apparent, though without quantitation, the designation as "significant" is premature. Also, do the authors have a lighter exposure of the blot, as it's possible this could be overexposed. In 4H (panel c vs d), the authors again claim from single cell images that Leu+Baf increases ICOSL levels. From this data, this claim is premature and should be quantified by FACS or confocal if it is meant to be a major point of the study.

b) Reviewer #3: Evidence for lysosomal degradation of ICOSL Figure 4 parts G and H. I found the argument with regard to the blocking of the lysosomal pathway by leupeptide and bafilomycin difficult to follow. The results Figure 4F show that with MCMV infection 70KDa ICOSL band is degraded however, blockage of the lysosomal pathway did not restore the band. Figure 4H while showing co-localization of ICOSL and m138 does not provide definitive evidence that this is in lysosomes. Taken together the definitive statement of redirection to lysomes for degradation is not fully supported. I think the results text could be edited to more clearly describe the experiments and the authors logic and the conclusion softened unless further evidence of lysosomal localisation and degradation is available.

3) Strengthen in vivo data. Where appropriate, ex vivo studies would be sufficient.

a) Reviewer #2: A significant assumption made by the authors is that ICOSL down-regulation occurs in DCs in vivo. ICOS-L expression by endothelial and epithelial cells has been described (e.g. doi:10.1073/pnas.092576699, doi:10.1002/eji.200535253). However, they provide no evidence for this, only bone marrow-derived cells and cell lines. The authors should demonstrate evidence for MCMV-induced down-regulation of ICOSL by primary mouse DCs (e.g. enriched primary splenic DCs, DC subsets in the peritoneum, or splenic DCs after high-dose systemic infection to enable detection of infected cells). Also, there needs to be discussion of the potential variation in infectivity of BM-DCs versus primary DCs (see PMID: 25120535).

b) Reviewer #2: The authors should strengthen the link between the impact on virus load in the salivary gland and m138 attenuation. The data regarding the impact of m138 on follicular T cell and antibody development is interesting and may influence control of virus reactivation, but it unlikely to explain the phenotype in the salivary gland given the apparent absence of antibody in control of primary MCMV infection. Indeed, it was been previously published that salivary gland IL-10+ CD4 T cell accumulation is ICOS-dependent in C57BL/6 mice (PMID: 27926930). Given that these cells are Th1 derived, it seems likely that m138 may promote virus persistence via antagonizing development of mucosal Th1 cell responses. The authors should therefore measure the impact of m138 attenuation on virus-specific CD4 T cells.

c) Reviewer #3 (Related to point b above): 2. It is clear that m138 is an important immune evasion protein with multiple functions and dissecting these in an in vivo model is challenging because of this. The experiments detailed in Figure 7 attempt to do this, however I found the description of the results and the logic to support the conclusions difficult to follow. In part this is because the wt MCMV and m138 mutants effect different immune evasion pathways relevant to NK Cells (m138 downregulates MULT1 and RAE1 but not H60 and T cells m138 downregulates CD80 and ICOSL). Monitoring late viral titres with these viruses, in addition to removing NK cells and NK cells with T cells adds to the complexity of the experiment. I think that an addition to the figure of a panel that details the effects on relevant molecules of the immune system by each virus mutant would be helpful and that the results text should be redrafted to more fully explain the observed results for each of the mutants.

---

## [Author Response]

Essential revisions:1) Strengthen Lysosomal degradation and ICOSL- m138 localisation:a) Reviewer #1: While the authors cite previous studies that colocalize LAMP1 and m138, those studies used different cell systems and did not actually quantify colocalization with Pearson coefficients. A key mechanistic argument of this study is m138-mediated targeting of ICOSL to lysosomes. It would seem appropriate for the authors to establish and quantify this colocalization in their specific system. (Additionally, the Thale 1994 paper referenced does not appear to match the statement its associated with.)

We agree with the reviewer that, being an important point in the manuscript, data regarding m138-mediated targeting of ICOSL to lysosomes need to be strengthen. To address this issue we have carried out several experiments. We have performed confocal microscopy studies on MCMV infected cells staining with the red fluorescence marker LysoTracker DND-99, and show now in Figure 3D, that m138 can be detected in lysosomal compartments during infection. Using the Coloc2 plugin from FIJI/ImageJ program we have quantified the m138/lysosome colocalization, obtaining Manders colocalization coefficients of M1=0.76 and M2=0.80 (indicated in the text). In addition, we have examined more precisely the colocalization of m138 and ICOSL in the infection context by confocal microscopy, in the presence and absence of the lysosomal inhibitors leupeptin and bafilomycin, and quantified the abundance of this cellular receptor in both conditions by flow cytometry. In panel B of the new Figure 5, which substitutes the original Figure 4H, we present confocal microscopy images illustrating that m138 and ICOSL colocalize. In addition, a stronger fluorescent signal corresponding to both ICOSL and m138 can be appreciated upon treatment with the lysosomal inhibitors. Moreover, in Figure 5C we show now the results of the quantification of the levels of ICOSL in permeabilized MCMV infected cells by flow cytometry, demonstrating that the expression of both ICOSL and m138 significantly increases after lysosomal inhibitor exposure.

As indicated, the Thale et al., 1994 reference after the statement mentioning earlier reports showing m138 localization in intracellular compartments in transfection assays has been removed.

b) Reviewer #2: Given the focus of DCs in this study, why did the authors use NIH 3T3 cells for the localizations studies? Some justification should be given and/or key observations should be repeated in these cell types.

The reason of using NIH3T3 cells for the colocalization studies was mainly due to the fact that these assays were done in parallel with the Western blot analysis, which allowed us to discern between intracellular ICOSL (the 51 kDa band) and ICOSL at the cell surface (the 70 kDa band). The absence of an appropriate antibody that permitted us to specifically detect ICOSL by Western blot required us to generate and employ a stable NIH3T3 HA-ICOSL cell line to perform the assays. However, taking into consideration the reviewer comment, we have now performed a new assay in DC2.4 cells to show that m138 also localizes in perinuclear compartments and cytoplasmic large punctate vesicles (Figure 3—figure supplement 1).

2) Clarify leupeptin/Bafilomycin blocking studies:a) Reviewer #1: In Figure 4G,4H, the authors claim that Leu+Baf results in "significant increase of the 51kDa band". An increase is apparent, though without quantitation, the designation as "significant" is premature. Also, do the authors have a lighter exposure of the blot, as it's possible this could be overexposed. In 4H (panel c vs d), the authors again claim from single cell images that Leu+Baf increases ICOSL levels. From this data, this claim is premature and should be quantified by FACS or confocal if it is meant to be a major point of the study.

We acknowledge the reviewer comment on the increase of the intracellular levels of ICOSL upon treatment with lysosomal inhibitors, which is related to issue 1a from reviewer #2 and that, as indicated above, we have now addressed obtaining results that more strongly support our conclusions.

Regarding Figure 5A (Figure 4G in the original manuscript), we proceeded to quantify the 51 kDa band for the different samples analyzed in the Western blot. A 1.5-fold increase on the levels of the 51 kDa ICOSL band was obtained after leupeptin and bafilomycin treatment in cells co-transfected with the m138-GFP protein, whereas the addition of the lysosomal inhibitors did not alter substantially (1.1-fold increase) that band in the cells co-transfected with the CTL-GFP protein. These values have been incorporated in the text corresponding to the figure in the revised manuscript.

Concerning the suggestion of presenting a lighter exposure of the Western blot in Figure 5A, we prefer to retain in the manuscript the exposure that we had in the original submitted version. This exposure permits to clearly visualize that the 70 kDa ICOSL band, which is observed in the absence of m138 (lanes 2 and 3) cannot be detected in the presence of the viral protein (lanes 4 and 5).

As indicated to reviewer #1 (in the response to comment 1a) we have examined more precisely the expression of ICOSL during MCMV infection in the presence and absence of the lysosomal inhibitors leupeptin and bafilomycin. On one hand, we used confocal microscopy and present now, in the new Figure 5B, images (with more than one cell per field) showing a more intense staining of ICOSL upon treatment with the lysosomal inhibitors. In addition, as suggested, we have quantified the abundance of ICOSL in MCMV infected cells under both conditions by flow cytometry, demonstrating in Figure 5C that the expression of ICOSL significantly increases upon lysosomal inhibitor exposure, from MFI values of 11.628 to 17.682, in NIH3T3 HA-ICOSL.

b) Reviewer #3: Evidence for lysosomal degradation of ICOSL Figure 4 parts G and H. I found the argument with regard to the blocking of the lysosomal pathway by leupeptide and bafilomycin difficult to follow. The results Figure 4F show that with MCMV infection 70KDa ICOSL band is degraded however, blockage of the lysosomal pathway did not restore the band. Figure 4H while showing co-localization of ICOSL and m138 does not provide definitive evidence that this is in lysosomes. Taken together the definitive statement of redirection to lysomes for degradation is not fully supported. I think the results text could be edited to more clearly describe the experiments and the authors logic and the conclusion softened unless further evidence of lysosomal localisation and degradation is available.

As indicated in previous answers to issues 1a to reviewer #1 and issue 2a to reviewer #2, we present in the revised version of the manuscript new data to more robustly support our conclusions related to the m138-induced degradation of ICOSL during MCMV infection. In addition, the corresponding text in this section has been slightly modified so it could be followed better.

3) Strengthen in vivo data. Where appropriate, ex vivo studies would be sufficient.a) Reviewer #2: A significant assumption made by the authors is that ICOSL down-regulation occurs in DCs in vivo. ICOS-L expression by endothelial and epithelial cells has been described (e.g. doi:10.1073/pnas.092576699, doi:10.1002/eji.200535253). However, they provide no evidence for this, only bone marrow-derived cells and cell lines. The authors should demonstrate evidence for MCMV-induced down-regulation of ICOSL by primary mouse DCs (e.g. enriched primary splenic DCs, DC subsets in the peritoneum, or splenic DCs after high-dose systemic infection to enable detection of infected cells). Also, there needs to be discussion of the potential variation in infectivity of BM-DCs versus primary DCs (see PMID: 25120535).

To address the reviewer comment we have conducted additional experiments in which we analyzed the downmodulation of ICOSL, as well as that of CD80, induced by MCMV on infected peritoneal DCs (CD11^+^ MHC II^+^ CD3^-^ CD19^-^ NKp46^-^ cells) of BALB/c infected mice. More importantly, to assess whether the effects were caused by m138, we also included in the assays infections with MCMV∆m138. The results, which have been incorporated in Figure 8B of the new version of the manuscript, show that MCMV downmodulates ICOSL and CD80 in these primary DCs during the in vivo infection, and confirmed the involvement of m138 in this process.

With regards to the potential variation in the infectivity ratio between BM-DCs and DCs in peritoneum, the amount of virus, number of DCs and time of infection need to be taken into account. Thus, for example, in our hands, when BM-DCs were infected in vitro with 3 PFU/cell for 24 hours, we usually got an infectivity rate of 15-20%. On the other hand, the number of DCs in the mouse peritoneum is low. In two individual experiments, we detected 15% of total DCs (live CD11c^+^ MHC II^+^ CD3^-^ CD19^-^ NKp46^-^ cells) among peritoneal exudate cells of uninfected mice (Author response image 1). This percentage dropped to 2-4% of total DCs when mice were given a high dose of virus (10^6^ PFU/mice) for 6 hours. Since in these conditions, DC infection rate varied between 5-15%, this means that less than 1% of peritoneal exudate cells corresponded to infected DCs. This number is in line with previous publications where MCMV-GFP virus was used (Hsu et al., 2009). Importantly, and despite these low numbers of infected DCs in mouse peritoneum, we managed to show the influence of the m138 protein on ICOSL and CD80 levels in infected DCs in vivo at early times post-infection (Figure 8B).

**Author response image 1. respfig1:** (A) Percentage (%) and (B) number (#) of total peritoneal DCs (live Lin^-^ CD11c^+^ MHC II^+^) and MCMV-infected DCs (m04^+^ cells from total DCs). BALB/c mice were i.p. infected with 10^6^ PFU/mice of MCMVwt and MCMVΔm138. DCs were isolated from the peritoneal cavity 6 hr post-infection and analyzed by flow cytometry. Results represent two merged individual experiments (mock n=8, MCMVwt n=6, MCMVΔ138 n=7). One-way ANOVA test was used for statistical analysis. Graphs show mean with SEM as error bars. ****, p ≤0.0001; ***, p ≤ 0.001; **, p ≤ 0.01; *, p ≤ 0.05.

b) Reviewer #2: The authors should strengthen the link between the impact on virus load in the salivary gland and m138 attenuation. The data regarding the impact of m138 on follicular T cell and antibody development is interesting and may influence control of virus reactivation, but it unlikely to explain the phenotype in the salivary gland given the apparent absence of antibody in control of primary MCMV infection. Indeed, it was been previously published that salivary gland IL-10+ CD4 T cell accumulation is ICOS-dependent in C57BL/6 mice (PMID: 27926930). Given that these cells are Th1 derived, it seems likely that m138 may promote virus persistence via antagonizing development of mucosal Th1 cell responses. The authors should therefore measure the impact of m138 attenuation on virus-specific CD4 T cells.

In response to the reviewer comment, we have now analyzed in the new version of the manuscript, CD4^+^ T cell responses to MCMV∆m138 in various organs of infected mice. We performed adoptive transfer assays of TCR-transgenic CD4^+^ T cells specific for the M25 protein (M25-II, Mandaric et al., PLoS Pathog 2012) into C57BL/6 SCID mice, NK cell-depleted, one day prior to infection with either MCMVwt or MCMV∆m138, and analyzed the expansion of CD4^+^ T cells (CD45.1^+^ CD4^+^) 7 days after infection in spleen, lungs and salivary glands. In this mouse model, after NK depletion, only endogenous mononuclear phagocytes are present, and thereby we can exclude the potential role in viral clearance of m138-dependent endogenous lymphocytes. The results of the assay, now included in the new Figure 8—figure supplement 1, show: i) the expansion of the transferred exogenous M25-specific CD4^+^ T cells upon antigen encounter (MCMV), and ii) that m138 contributes to diminish the proliferation of these virus-specific CD4^+^ T cells in spleen and lungs of infected mice, but not in the salivary gland.

Although the data argue against a role of m138 in evasion of CD4^+^ T cells in salivary glands, it must be taken in account that we have only analyzed CD4^+^ T cell responses specific to the M25 epitope, and that m138 might be influencing in a different way CD4^+^T cell responses specific for other MCMV epitopes. In addition, since CD4^+^ T cells (and not CD8^+^ T cells) are crucial for the control of MCMV infection in the salivary glands, data could indicate that CD4^+^ T cells are more efficient in virus elimination in MCMVΔm138- than in MCMVwt-infected mice, at early days post-infection. This will result in less virus in the salivary glands of MCMVΔm138-infected mice at 7 days p.i. As M25-specific CD4^+^ T cell proliferation in salivary glands is dependent on antigen load, this could explain why there are no differences in the percentage of M25-specific CD4^+^ T cells between the two groups of infected mice.

The text describing the results of Figure 8—figure supplement 1 and the corresponding discussion have been now included in the modified version of the manuscript.

c) Reviewer #3 (Related to point b above): 2. It is clear that m138 is an important immune evasion protein with multiple functions and dissecting these in an in vivo model is challenging because of this. The experiments detailed in Figure 7 attempt to do this, however I found the description of the results and the logic to support the conclusions difficult to follow. In part this is because the wt MCMV and m138 mutants effect different immune evasion pathways relevant to NK Cells (m138 downregulates MULT1 and RAE1 but not H60 and T cells m138 downregulates CD80 and ICOSL). Monitoring late viral titres with these viruses, in addition to removing NK cells and NK cells with T cells adds to the complexity of the experiment. I think that an addition to the figure of a panel that details the effects on relevant molecules of the immune system by each virus mutant would be helpful and that the results text should be redrafted to more fully explain the observed results for each of the mutants.

As required by the reviewer we have introduced a panel in Figure 8 detailing the effects on the molecules of the immune system targeted by each MCMV mutant. The text corresponding to the results obtained in the in vivo assay with these MCMV mutants has been redrafted too.